   

EMBO
Molecular Medicine

# ERK5 suppression overcomes FAK inhibitor resistance in mutant KRAS-driven non-small cell lung cancer

Chiara Pozzato [ID][1], Gonçalo Outeiro-Pinho [ID][1], Mirco Galiè[2], Giorgio Ramadori [ID][3,4] & Georgia Konstantinidou [ID][1✉]

## Abstract

**Mutated KRAS serves as the oncogenic driver in 30% of non-small cell lung cancers (NSCLCs) and is associated with metastatic and therapy-resistant tumors. Focal Adhesion Kinase (FAK) acts as a mediator in sustaining KRAS-driven lung tumors, and although FAK inhibitors are currently undergoing clinical development, clinical data indicated that their efficacy in producing long-term anti-tumor responses is limited. Here we revealed two FAK interactors, extracellular-signal-regulated kinase 5 (ERK5) and cyclin-dependent kinase 5 (CDK5), as key players underlying FAK-mediated maintenance of KRAS mutant NSCLC. Inhibition of ERK5 and CDK5 synergistically suppressed FAK function, decreased proliferation and induced apoptosis owing to exacerbated ROS-induced DNA damage. Accordingly, concomitant pharmacological inhibition of ERK5 and CDK5 in a mouse model of Kras[G12D]-driven lung adenocarcinoma suppressed tumor progression and promoted cancer cell death. Cancer cells resistant to FAK inhibitors showed enhanced ERK5-FAK signaling dampening DNA damage. Notably, ERK5 inhibition prevented the development of resistance to FAK inhibitors, significantly enhancing the efficacy of anti-tumor responses. Therefore, we propose ERK5 inhibition as a potential co-targeting strategy to counteract FAK inhibitor resistance in NSCLC.**

**Keywords** Lung Cancer; Focal Adhesions; Drug Resistance; Combination Therapy; FAK and ERK5 Inhibitors
**Subject Categories** Cancer; Respiratory System

## Introduction

Non-small cell lung cancer (NSCLC) is the second most diagnosed cancer and the leading cause of cancer-related deaths (Siegel et al, 2021). About 30% of NSCLCs are driven by an activating *KRAS* mutation, which renders them aggressive and treatment resistant. Over the past years, there has been an increase in innovative targeted therapies, with the goal of improving the treatment management of patients with NSCLC. Notably, sotorasib and adagrasib have gained FDA approval for targeting KRAS[G12C] in patients with advanced NSCLC, providing a significant step forward in cancer therapy (Lanman et al, 2020). Despite these advancements, the effectiveness of these therapies remains limited, primarily attributed to drug resistance, a persistent hurdle in the landscape of targeted cancer treatments (Mohanty et al, 2023). Therefore, the identification of mechanisms by which cancer cells evade treatment in mutant KRAS-driven tumors is of paramount importance.

FAK (gene name *PTK2*) is a non-receptor tyrosine kinase that plays a role in regulating multiple cancer cellular functions, such as cell proliferation, migration, invasion and metastasis (Diaz Osterman et al, 2019; Lee et al, 2015; Mitra et al, 2005; Skinner et al, 2016). Accordingly, FAK is found commonly overexpressed in invasive and metastatic cancers (Weiner et al, 1993), supporting its targeting as a valuable cancer therapeutic strategy.

We previously found that FAK is required for the maintenance of *KRAS* mutant NSCLC. FAK inhibition in genetically engineered KRAS-driven mouse cancer models led to the regression of lung adenocarcinomas (Konstantinidou et al, 2013). These findings, set the basis for a phase-II clinical trial with single-agent defactinib (VS-6063), an orally available ATP-competitive FAK inhibitor, in 55 heavily pretreated *KRAS* mutant NSCLC patients (clinical trial identifier: NCT01951690) (Gerber et al, 2020). The results of the trial were promising as defactinib provided a disease control rate (DCR) of ~50% (DCR is the % of patients whose tumor shrinks or remains stable over a certain time period), suggesting a cytostatic effect. However, compared to the efficacy observed in preclinical trials of lung cancer in mice, these results were overall underwhelming. This is also in line with previous findings, which demonstrated that the inhibition of FAK in human xenograft models of lung cancer primarily yielded a cytostatic effect (Konstantinidou et al, 2013). These observations hint at the rapid activation of resistance mechanisms in human tumors following FAK inhibition. The identification of these mechanisms will

[1]Institute of Pharmacology, University of Bern, 3010 Bern, Switzerland. [2]Department of Neuroscience, Biomedicine and Movement, University of Verona, 37134 Verona, Italy. [3]Department of Cell Physiology and Metabolism, University of Geneva, 1211 Geneva, Switzerland. [4]Diabetes Center of the Faculty of Medicine, University of Geneva, 1211 Geneva, Switzerland. ✉E-mail: georgia.konstantinidou@unibe.ch

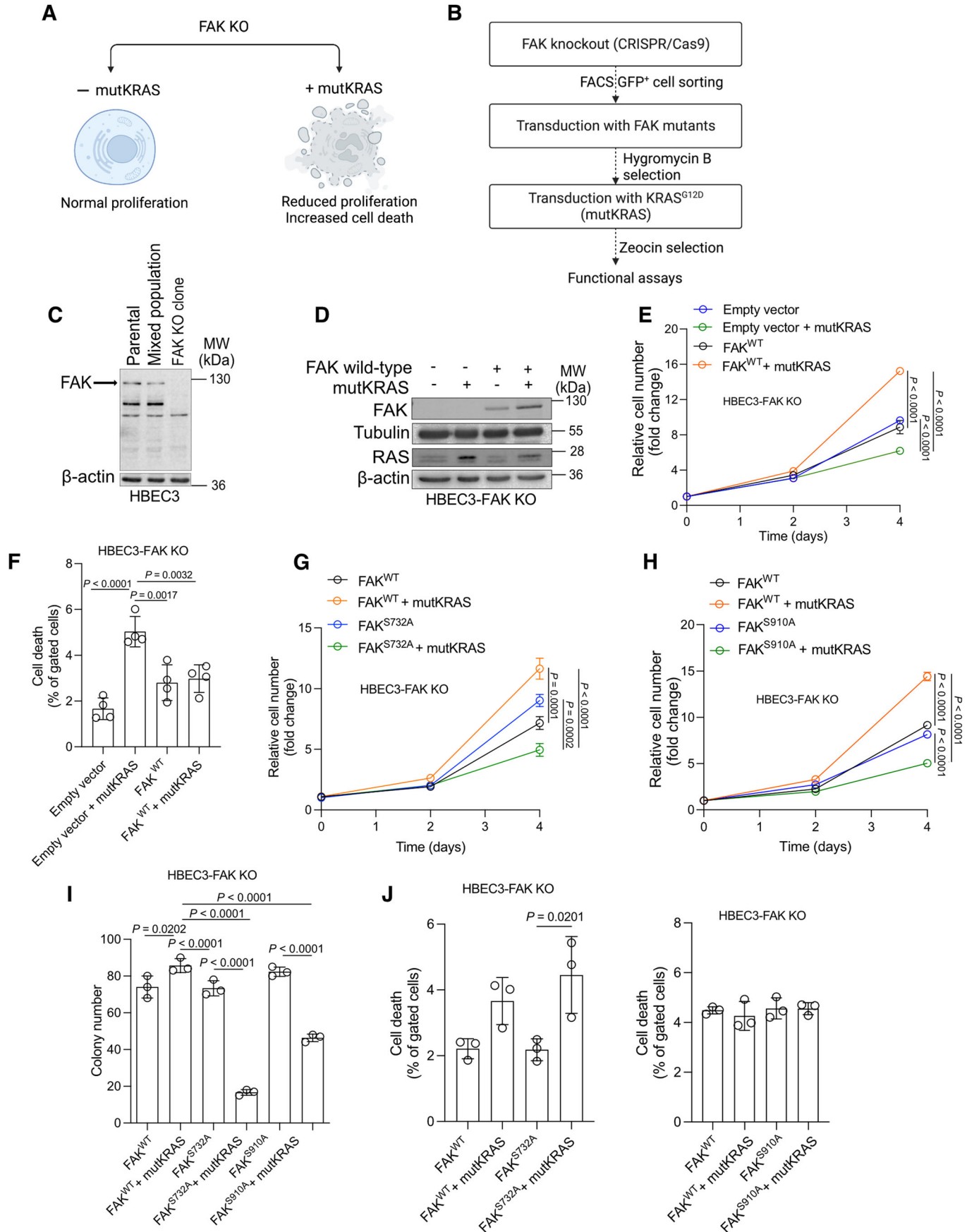

◀  **Figure 1.   ERK5 and CDK5 are key regulators of FAK function in mutant KRAS NSCLC.**

(A) Scheme depicting the critical dependency of KRAS mutant cancer cells on FAK. (B) Scheme showing the stepwise preparation of the HBEC3 cellular model. (C) Immunoblot against FAK in HBEC3 parental, HBEC3 mixed population before GFP⁺ cell FACS sorting and HBEC3 FAK knockout clone. (D) Immunoblot analysis in HBEC3-FAK KO cell line transduced either with empty vector (pWZL-Hygro) or wild-type FAK (pWZL-Hygro-FAK WT) in the absence or presence of mutKRAS$^{G12D}$ (pBABE-zeo or pBABE-zeo-KRAS $^{G12D}$); mutKRAS: mutant KRAS. (E) Relative cell number of HBEC3-FAK KO cell line transduced as in (D); $n = 3$. (F) Relative quantification of cell death by flow cytometry analysis of Annexin V-Atto 633 (AV) -positive + Annexin V/PI (AV/PI)-positive + PI-positive gated populations in HBEC3-FAK KO cells previously transduced as in (D); $n = 4$. (G) Relative cell number of HBEC3-FAK KO cell line transduced either with wild-type FAK (pWZL Hygro-FAK WT) or FAK$^{S732A}$ (pWZL-Hygro-FAK$^{S732A}$) in the absence or presence of mutant KRAS$^{G12D}$ (pBABE-zeo or pBABE-zeo-KRAS $^{G12D}$). Mutant KRAS: mutKRAS; $n = 3$. (H) Relative cell number of HBEC3-FAK KO cell line transduced either with wild-type FAK (pWZL Hygro-FAK WT) or FAK$^{S910A}$ (pWZL-Hygro-FAK$^{S910A}$) in the absence or presence of mutKRAS$^{G12D}$ (pBABE-zeo or pBABE-zeo-KRAS $^{G12D}$). Mutant KRAS: mutKRAS; $n = 3$. (I) Relative quantification of colony forming capacity of HBEC3-FAK KO cells previously transduced as indicated; $n = 3$. (J) Quantification of cell death by flow cytometry analysis of Annexin V-Atto 633 (AV)-positive + Annexin V/PI (AV/PI)-positive + PI-positive HBEC3-FAK KO cells previously transduced as indicated; $n = 3$. Graphical data are mean ± SD. Statistical analyses were done using one-way ANOVA; $n$, number of biologically independent samples. Source data are available online for this figure.

provide information about the required feedback loops leading to FAK inhibitor resistance.

The autophosphorylation of FAK at Y397 is a key step for its activation as it provides a binding site for Src kinase and, consequently, phosphorylation at Y576 and Y577 in the activation loop of its kinase domain (Lietha et al, 2007). However, FAK is subjected to multiple other phosphorylation events that, based on the context, are important for FAK activity. Indeed, given its central role in focal adhesions and crosstalk with the extracellular matrix, FAK is regulated by multiple pathways, including growth factor receptor-bound protein 2/son of sevenless (Grb2/SOS), mitogen-activated protein kinase 7 (MAPK7/ERK5) and cyclin-dependent kinase 5 (CDK5) (Schlaepfer et al, 1994; Villa-Moruzzi, 2007; Xie et al, 2003). Of note, the possible contribution of the different regulatory phosphorylation sites on FAK during lung cancer progression and development of drug resistance has not yet been explored.

Here, we set to identify possible FAK inhibitor resistance mechanisms in NSCLC by considering essential interacting proteins for FAK-mediated lung tumor maintenance. Understanding these mechanisms may provide ways for overcoming FAK drug resistance and greatly improve cancer treatment.

## Results

### Serines 732 and 910 of FAK are critical for the proliferation of mutant KRAS-transformed lung cells

Genetic or pharmacological inhibition of FAK leads to reduced proliferation and increased cell death in KRAS mutant lung cancer cells, while sparing cells carrying wild-type KRAS (Konstantinidou et al, 2013) (Fig. 1A). Therefore, we took advantage of this vulnerability to dissect the specific regulatory site/s of FAK required for the maintenance of KRAS mutated cancer cells. To establish the cellular model, first we knocked out FAK via Clustered Regularly Interspaced Short Palindromic Repeats (CRISPR)/Cas9 and knocked down p53 in immortalized bronchial epithelial cells, HBEC3-KT (thereafter HBEC3-FAK KO) (Figs. 1B,C and EV1A) (Ramirez et al, 2004). As expected, HBEC3-FAK KO cells displayed similar proliferation with the parental control HBEC3 cells (Fig. EV1B). However, ectopic mutant KRAS (mutKRAS) expression in HBEC3-FAK KO cells reduced proliferation and triggered cell death, while re-expression of wild-type FAK (FAK$^{WT}$) rescued cell proliferation and dampened cell death (Fig. 1D–F). Contrary, expression of a loss-of-function point mutant of FAK at position 397 (FAK$^{Y397F}$), a phosphorylation site which is essential for FAK activity, only partially rescued cell proliferation and triggered a similar suppression of colony forming capacity compared to the empty vector control in presence of mutKRAS (Fig. EV1C–E).

Next, to identify regulatory sites on FAK that are essential for lung tumor maintenance, we transduced HBEC3-FAK KO cells with vectors expressing human FAK carrying 7 distinct loss-of-function point mutations that disrupt the phosphorylation sites of known binding partners, namely p130Cas (P712/713A) (Pylayeva et al, 2009), GSK3β/PP1 (S722A) (Bianchi et al, 2005), Src (Y861F) (Schaller et al, 1994), Grb2/SOS (Y925F) (Schlaepfer et al, 1994), CDK5 (S732A) (Xie et al, 2003), ERK5 (S910A) (Villa-Moruzzi, 2007) or compromising FAK's nuclear localization signal (R177/178A) (Ossovskaya et al, 2008) (Figs. 1B and EV1F) and assessed their impact on cell proliferation in the presence or absence of mutKRAS. Whereas introduction of FAK$^{R177/178A}$, FAK$^{P712/713A}$, FAK$^{S722A}$, FAK$^{Y861F}$ and FAK$^{Y925F}$ allowed a similar proliferation of HBEC3-FAK KO cells in presence or absence of mutKRAS, FAK$^{S732A}$ and FAK$^{S910A}$ evidenced impaired proliferation only in the presence of mutKRAS (Figs. 1G,H and EV1G–J). Notably, both FAK$^{S732A}$ and FAK$^{S910A}$ impaired colony formation, yet only FAK$^{S732A}$ triggered cell death in presence of mutKRAS (Fig. 1I,J). Altogether, these data suggest that the phosphorylation of FAK at Serines 732 and 910 promotes cancer cell proliferation and colony forming capacity.

### Co-inhibition of ERK5 and CDK5 synergistically suppresses FAK function, proliferation and survival of NSCLC cells

FAK-S732 and FAK-S910 are sites that are phosphorylated by CDK5 and ERK5, respectively (Jiang et al, 2020b; Villa-Moruzzi, 2007; Xie et al, 2003), and our data suggest that these two factors are important for FAK's function in mediating tumorigenesis of NSCLC cancer cells bearing mutKRAS. To further test this hypothesis, we assessed the effect of pharmacological inhibition of ERK5 (XMD8-92) (Yang et al, 2010) and CDK5 (Seliciclib) (Cicenas et al, 2015) in a panel of NSCLC cancer cells (A549, A427 and H460) bearing mutKRAS. Our data evidenced that while the treatment with XMD8-92 or Seliciclib alone failed to substantially decrease proliferation, the combination of both inhibitors synergistically suppressed cell proliferation (Fig. 2A). Furthermore,

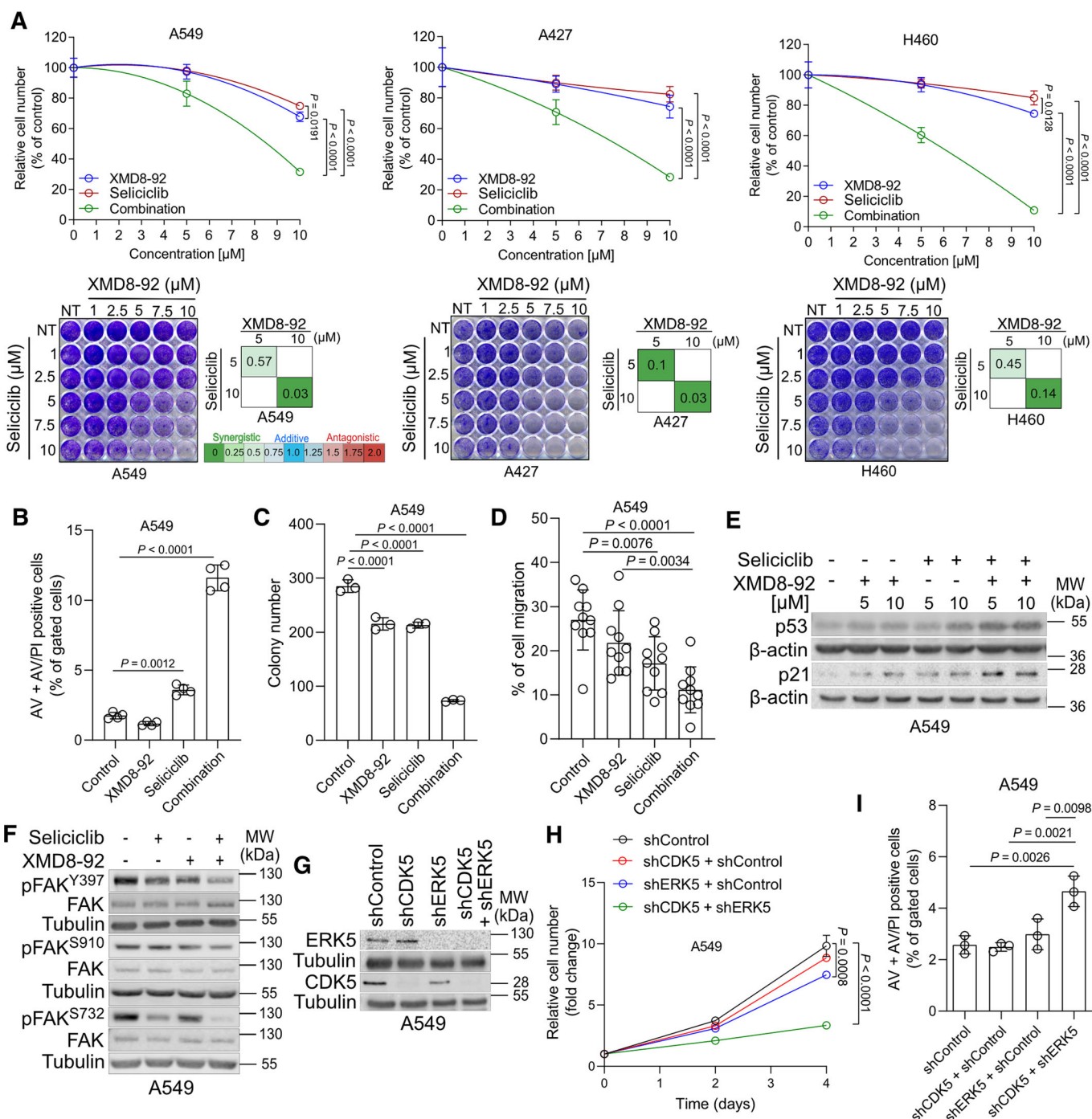

**Figure 2. ERK5 and CDK5 synergistically promote FAK function.**

(A) Percentage of proliferation inhibition of A549, A427 and H460 cell lines treated with increasing doses of XMD8-92 or Seliciclib or in combination (top) 48 h after treatment and representative crystal violet-stained cells 72 h after drug treatment (bottom). The combination index (CI) showing the synergistic effect of combination of the 2 drugs is indicated; $n = 3$. (B–D) Relative quantification of Annexin V (AV) + Annexin V/PI (AV/PI)-positive cells by flow cytometry; $n = 4$ (B), colony forming capacity; $n = 3$ (C) and percentage of cell migration; $n = 10$ (D) of A549 cells treated with 10 μM XMD8-92 or Seliciclib or in combination, except for the colony formation assay in which cells were treated with 5 μM of each drug. (E) Immunoblot analysis of the indicated targets in A549 cells treated with the indicated doses of Seliciclib or XMD8-92 or in combination for 24 h. (F) Immunoblot analysis of the indicated targets in A549 cells treated with 10 μM XMD8-92 or 10 μM Seliciclib or in combination for 24 h. (G) Immunoblot analysis for the indicated targets in A549 cells transduced with shRNA control (pLKO.1 hygro + Tet-pLKO-puro) or a shRNA against CDK5 (Tet-pLKO-puro-shCDK5 + pLKO.1 hygro), or a shRNA against ERK5 (pLKO.1 hygro-shERK5 + Tet-pLKO-puro) or in combination (Tet-pLKO-puro-shCDK5 + pLKO.1 hygro-shERK5). After transduction and selection, cells were harvested for protein extraction 72 h after doxycycline (1 μg/mL) induction. (H, I) Relative cell number (H) and Annexin V (AV) + Annexin V/PI (AV/PI)-positive cell quantification by flow cytometry (I) in A549 cells treated as in (G) 72 h after doxycycline (1 μg/mL) induction; $n = 3$. Graphical data are mean ± SD. Statistical analyses were done using one-way ANOVA; $n$, number of biologically independent samples. Source data are available online for this figure.

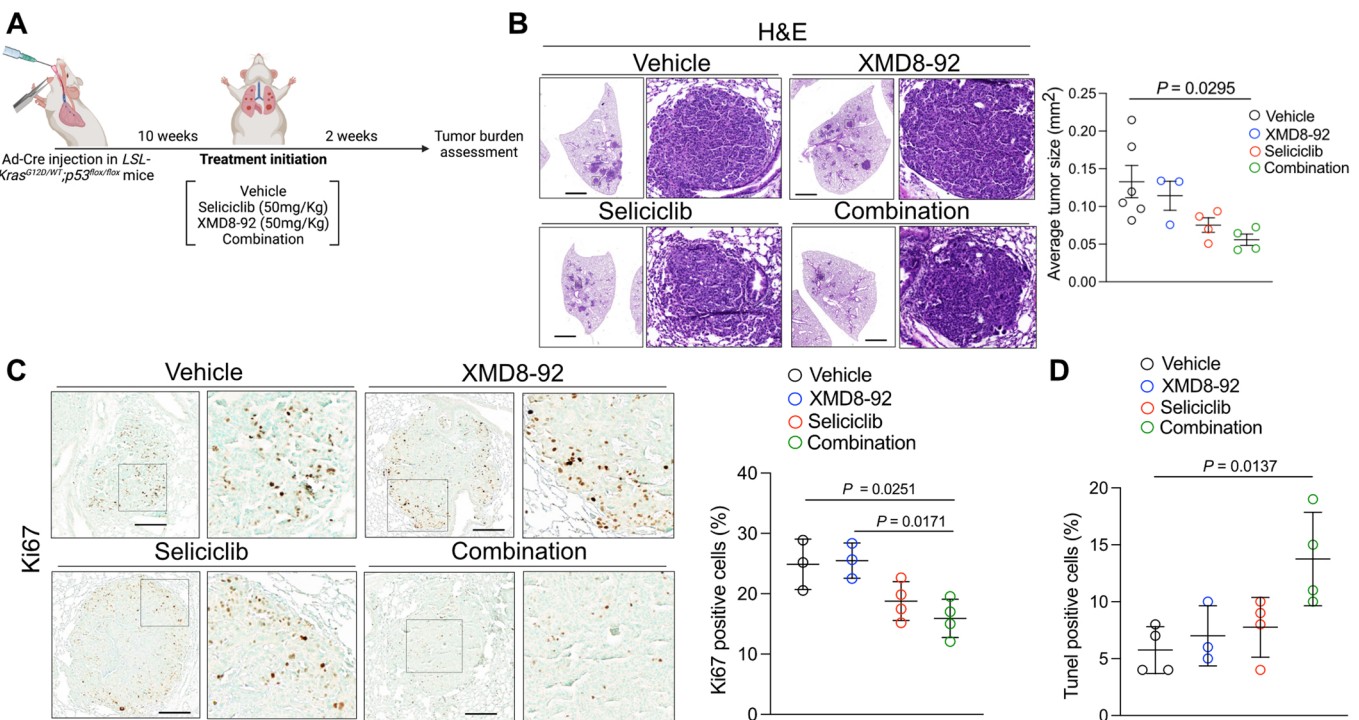

**Figure 3. Co-inhibition of ERK5/CDK5 suppresses lung tumorigenesis.**

(A) Representative scheme of the in vivo experiment workflow. (B) Representative hematoxylin & eosin (H&E) staining (left) and quantification of the average tumor size (right) of lung tissue from *LSL-Kras*$^{G12D/WT}$*;p53*$^{flox/flox}$ mice 10 weeks after Cre induction and after 2 weeks of treatment with vehicle, XMD8-92 (50 mg/Kg), Seliciclib (50 mg/Kg) or combination. Scale bar: 1 mm; *n* mice/group: 6, 3, 4, 4. Graphical data are ± SEM. (C) Representative images of immunohistochemistry against Ki67 (left) and quantification of Ki67-positive cells (right) in lung tissue from *LSL-Kras*$^{G12D/WT}$*;p53*$^{flox/flox}$ mice, treated as in (B). Scale bar: 100 μm; *n* mice/group: 3, 3, 4, 4. Graphical data are mean ± SD. (D) Tunel-positive cell quantification in lung tissue from *LSL-Kras*$^{G12D/WT}$*;p53*$^{flox/flox}$ mice, treated as in (B); *n* mice/group: 4, 3, 4, 4. Graphical data are mean ± SD. Statistical analyses were done using one-way ANOVA; *n*, number of biologically independent samples. Source data are available online for this figure.

concurrent inhibition of ERK5 and CDK5 evidenced a synergistic induction of cell death by apoptosis as shown by the increase in Annexin V and propidium iodide (ANN/PI)-positive cells, suppression of colony forming capacity and cell migration in A549 and A427 cells (Figs. 2B–D and EV2). Lastly, concurrent inhibition of ERK5 and CDK5 increased both p53 and p21 protein levels (Fig. 2E), supporting the anti-proliferative and pro-apoptotic impact of the dual inhibition and recapitulating previous observations showing an inhibitory role of FAK on p53 (Lim et al, 2008).

Immunoblot in A549 lung cancer cells evidenced that concurrent inhibition of ERK5 and CDK5 cooperated in suppressing FAK activity as evidenced by the decreased phosphorylation of both FAK$^{S910}$ and FAK$^{S732}$ as well as FAK$^{Tyr397}$, which is essential for FAK full activation (Fig. 2F). These data suggest an interplay between ERK5 and CDK5 in sustaining FAK activity in NSCLC cells.

XMD8-92 has reported activity against several other targets including the bromodomain family member (BRD4) and leucine-rich repeat kinase 2 (LRRK2) (Yang et al, 2010), while seliciclib is also active against CDK1, CDK2, CDK7 and CDK9 (Cicenas et al, 2015). Moreover, in some experimental settings, ERK5 inhibitors failed to recapitulate ERK5 genetic ablation phenotypes (Lochhead et al, 2020). To rule out whether the synergistic anti-proliferative and pro-apoptotic effects we observed with the compounds were specific to ERK5 and CDK5 inhibition, we repeated the above experiments by employing small hairpin RNA (shRNA)-mediated

knock down (Fig. 2G). We found that knockdown of ERK5 and CDK5 synergize in suppressing cancer cell proliferation and induce apoptosis of A549 cells only when combined, confirming a role of ERK5 and CDK5 in promoting the proliferation and survival of NSCLC cells (Fig. 2H,I).

Taken together, these results suggest that ERK5 and CDK5 are critical for FAK function in NSCLC cells carrying mutKRAS.

## Pharmacological inhibition of ERK5 and CDK5 suppresses tumor progression in a mouse model of Kras$^{G12D}$-driven lung adenocarcinoma

Our in vitro results suggest that concurrent inhibition of ERK5 and CDK5 synergistically decrease cell proliferation and increase cell death in mutKRAS NSCLC cell lines. To assess the impact of concurrent inhibition of ERK5 and CDK5 in vivo, we generated mice carrying a Cre-activatable *Kras*$^{G12D}$ allele (*LSL-Kras*$^{G12D}$) and a *p53* conditional knockout allele, *p53*$^{flox/flox}$ (named *LSL-Kras*$^{G12D/WT}$; *p53*$^{flox/flox}$, thereafter KP). Adenovirus-mediated Cre delivery to the lungs, results in expression of a Kras$^{G12D}$ allele and concomitant p53 deletion (*Kras*$^{G12D/WT}$; *p53*$^{-/-}$), leading to the development of lung adenocarcinomas (DuPage et al, 2009; Jackson et al, 2001). 10 weeks after Cre delivery, mice were treated with XMD8-92 or Seliciclib or in combination for 2 weeks (Fig. 3A). Both compounds were well tolerated either when given alone or in combination (Fig. EV3).

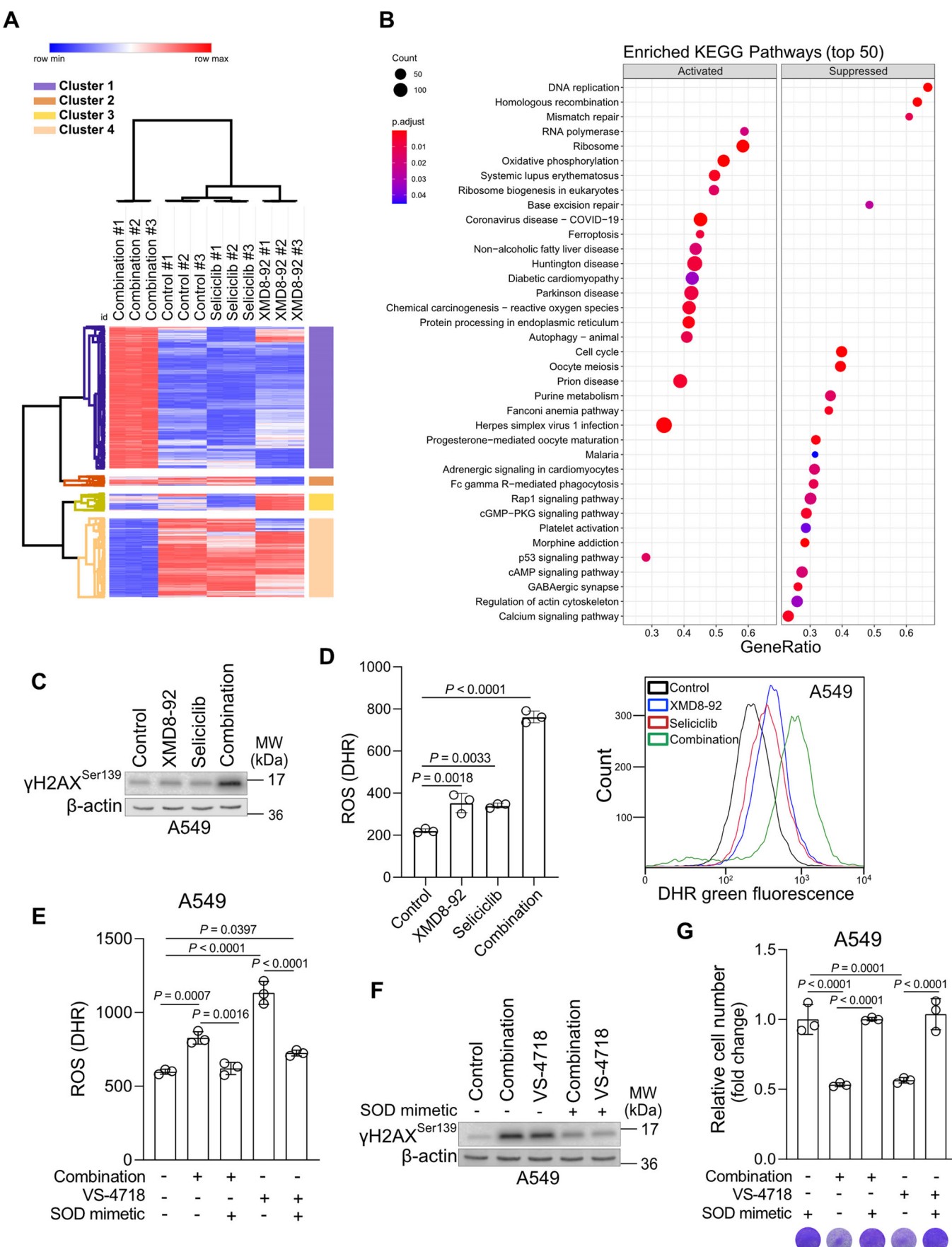

**Figure 4.   Co-inhibition of ERK5/CDK5 or FAK triggers ROS-induced DNA damage.**

(**A**) Hierarchical clustering (Pearson Correlation, average linkage) of genes with standard deviation at top 5% showing a clear separation between A549 cells treated with DMSO (Control) or XMD8-92 or Seliciclib or combination of the two drugs (10 μM each) for 12 h; $n = 3$. (**B**) Dotplot showing the results of KEGG pathway enrichment analysis of genes that are activated or suppressed in A549 treated with XMD8-92 and Seliciclib in combination (10 μM). (**C**) Immunoblot analysis of the indicated targets in A549 cells treated for 12 h with XMD8-92 and Seliciclib (10 μM for each drug) alone or in combination. (**D**) Quantification of DHR (ROS marker, green) (left) and representative flow cytometry histogram (right) of A549 cells treated as in (**C**); $n = 3$. (**E**) Quantification of DHR (ROS marker, green) in A549 cells treated with the combination of XMD8-92 and Seliciclib (10 μM) or VS-4718 (5 μM) in the presence or absence of the SOD mimetic, MnTMPyP (25 μM); $n = 3$. (**F**) Immunoblot analysis of the indicated targets in A549 cell line treated as in (**E**) except VS-4718: 2.5 μM. (**G**) Relative cell number (top) and representative crystal violet images of A549 cell line treated as in (**F**) for 96 h; $n = 3$. Graphical data are mean ± SD. Statistical analyses were done using one-way ANOVA; $n$, number of biologically independent samples. Source data are available online for this figure.

Tumor burden quantification revealed that concurrent ERK5 and CDK5 inhibition cooperated in suppressing lung tumor burden compared to vehicle or single treatment cohorts (Fig. 3B). The anti-tumor effect of the concurrent ERK5 and CDK5 inhibition was due to decreased proliferation as shown by the decrease in Ki-67, a cell proliferation marker and increased cell death as shown by the increased number of Tunel-positive cells, a marker of apoptosis (Fig. 3C,D).

## ERK5 and CDK5 inhibition increases intracellular reactive oxygen species levels causing DNA damage in NSCLC cells

To reveal the molecular mechanisms mediating the synergistic effect of ERK5 and CDK5 inhibition, we performed RNA sequencing analysis in A549 cells treated with XMD8-92 and Seliciclib either alone or in combination. The expression profile of the genes at the top 5% of standard deviation (SD) revealed a clearcut separation between A549 cells treated with XMD8-92 and Seliciclib and the single treatments or control group (Fig. 4A). Kyoto encyclopedia of genes and genomes (KEGG) pathway enrichment and gene set enrichment analysis (GSEA) showed that the combination drug treatment altered molecular pathways involved in DNA repair, cell cycle progression, oxidative phosphorylation (OXPHOS) and p53 signaling (Fig. 4B). Notably, the signaling pathways that were found to be altered by concomitant ERK5 and CDK5 inhibition have been previously reported to be affected by FAK suppression, confirming that ERK5 and CDK5 inhibition recapitulate, at least in part, FAK inhibition (Pylayeva et al, 2009; Zhang et al, 2016). For instance, FAK inhibition has been previously shown to trigger an increase in DNA damage (Tang et al, 2016). Indeed, we confirmed that concomitant ERK5 and CDK5 inhibition synergistically induced DNA damage as shown by the increased protein levels of γ-H2AX, a DNA damage marker (Sharma et al, 2012) (Fig. 4C).

KEGG pathway enrichment analysis provided evidence to support an increase in reactive oxygen species (ROS) production upon combinatorial ERK5 and CDK5 inhibition. Indeed, co-inhibition of ERK5 and CDK5 in A549 cells led to a synergistic increase in ROS production by about 2-fold compared to the single drug treatments and 4-fold compared to the control (Fig. 4D). Interestingly, a similar ROS increase was observed by direct inhibition of FAK autophosphorylation with VS-4718 or FAK kinase activity with the ATP-competitive inhibitor, PF-562271 (Figs. 4E and EV4A). Notably, the ROS triggered by the 2 different FAK inhibitors or by the combination of ERK5 and CDK5 inhibitors could be rescued by treatment with MnTMPyP, a cell-permeable superoxide dismutase (SOD) mimetic (Figs. 4E and EV4A).

High cellular ROS levels cause irreversible damage to proteins, DNA and lipids (Juan et al, 2021; Srinivas et al, 2019). To assess whether the high ROS levels were responsible for the increased DNA damage upon concurrent inhibition of ERK5 and CDK5 or direct FAK inhibition, we treated cells with the MnTMPyP to quench ROS (Liang et al, 2009). The pretreatment of A549 cells with MnTMPyP was sufficient to rescue DNA damage upon concurrent ERK5 and CDK5 or FAK inhibition (Figs. 4F and EV4B) Moreover, MnTMPyP treatment resulted in a complete recovery of the proliferative capacity of cancer cells treated with a combination of ERK5 and CDK5 or direct FAK inhibition (Fig. 4G). Taken together, our results suggest that similarly to FAK inhibition, co-inhibition of ERK5 and CDK5 trigger ROS-induced DNA damage in NSCLC cells.

## FAK inhibitor resistance is triggered by compensatory upregulation of ERK5

To establish an in vitro model mimicking FAK inhibitor resistance in NSCLC, we treated A549 cells with increasing doses of VS-4718, which resulted in the enrichment of drug-resistant cells that were characterized by slow proliferation and a mesenchymal-like morphology (Fig. 5A,B). Interestingly, the mesenchymal-like morphology was reversible upon VS-4718 withdrawal, pointing to a drug-tolerant cancer cell state (thereafter named VS4718-T) (Fig. 5A).

To further characterize the mechanisms of FAK inhibitor tolerant cancer cells, we performed RNA sequencing either on A549 parental cells, or VS4718-T or from cells acutely treated with VS-4718. Hierarchical clustering of the genes at the top 5% SD revealed a clear separation of differentially expressed genes in VS4718-T group compared to control or acutely treated with VS-4718 for 12 h (Fig. 5C). Hierarchical clustering identified 5 major clusters out of which 2 were highly abundant (cluster 1 and 5). Analysis of functional categories via Metascape (Zhou et al, 2019) evidenced that the genes that belonged to cluster 1 (downregulated in VS4718-T group) were associated with DNA replication, cell cycle and DNA repair, while the genes of cluster 5 (upregulated in VS4718-T group) were representing genes predominantly related to cell adhesion and epithelial-mesenchymal transition (EMT) (Fig. EV4C). Notably, TRRUST (Han et al, 2018) analysis revealed that the cluster 5 was enriched in targets of many transcription factors that, according to STRING database (https://string-db.org), had been previously shown to interact in multiprotein complexes. These included the signal transducer and activator of transcription 3 (STAT3) and TWIST1, both major mediators of EMT and drug resistance (Lee et al, 2014) (Fig. 5D). Moreover, the analysis of RNA-seq data showed down-regulation of epithelial markers and

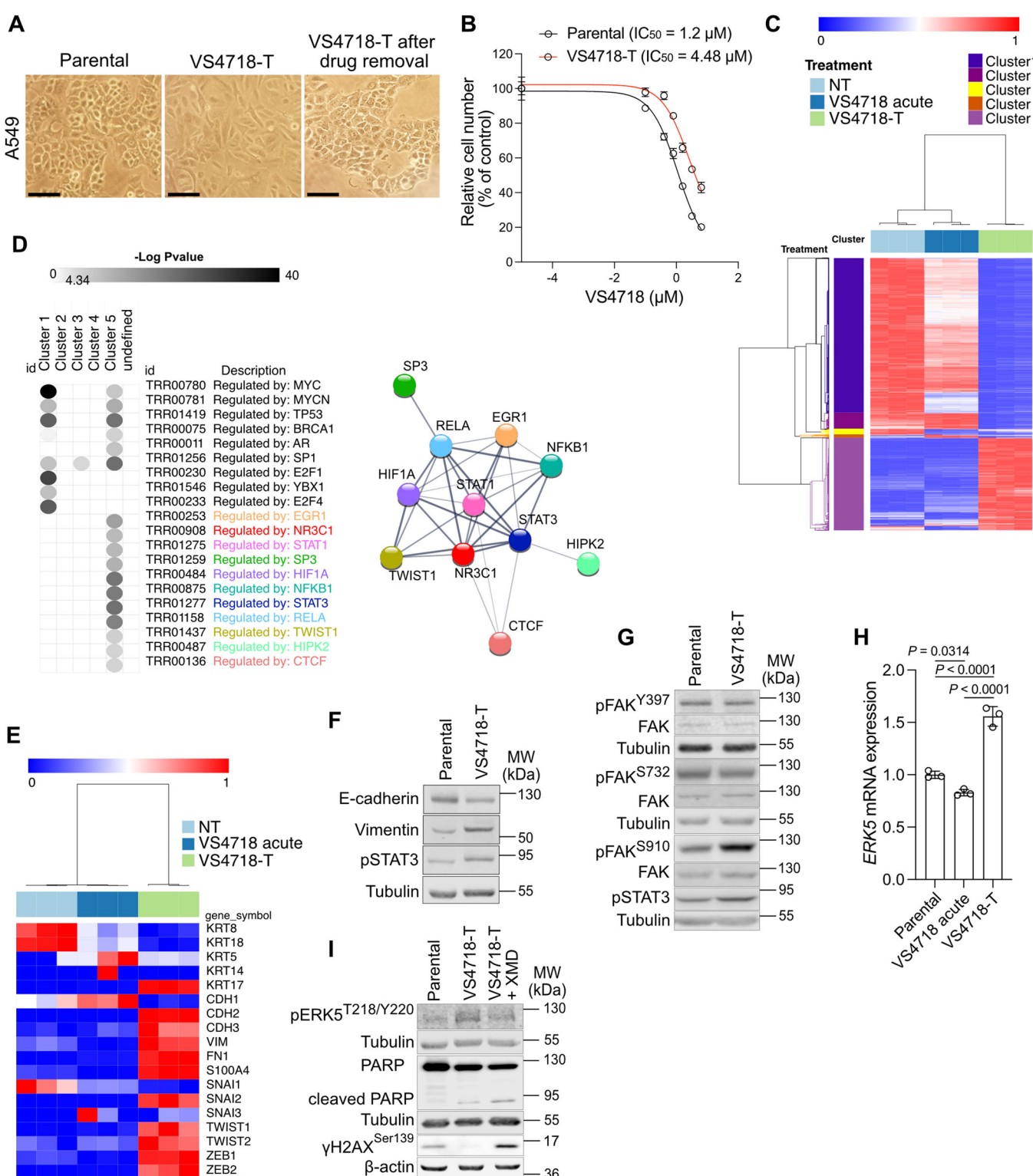

upregulation of EMT markers in VS4718-T cells (Fig. 5E). Accordingly, immunoblot evidenced a decrease in E-cadherin, which was concomitant with an increase in vimentin protein levels as well as pSTAT3, confirming that prolonged FAK inhibition results in STAT3 hyperactivation and EMT (Fig. 5F).

ERK5 and CDK5 aside from mediating the activation of FAK, control multiple other signaling pathways, including STAT3 (Giurisato et al, 2018; Hwang and Namgung, 2021; Stecca and Rovida, 2019). Therefore, we assessed whether ERK5 and/or CDK5 were responsible for FAK inhibitor-induced tolerance. Immunoblot

◄

**Figure 5.  ERK5 feedback activation induces FAK inhibitor tolerant cancer cells by suppressing DNA damage and cancer cell death.**

(**A**) Representative bright-field microscopy images of parental vehicle-treated A549 cells (left), VS-4718 tolerant cells (VS4718-T, middle) and VS4718-T upon withdrawal of the drug for 48 h (right). To obtain the VS-4718 tolerant (VS4718-T) cells, parental cells were treated with increasing doses of VS-4718 for 4 weeks and were after that maintained in 2.5 μM of VS-4718. Scale bars: 100 μm. (**B**) Percentage of proliferation inhibition of parental and VS4718-T A549 cells treated with increasing doses of VS-4718. Cell proliferation was determined 72 h post-treatment; $n = 3$. (**C**) Hierarchical clustering (Pearson Correlation, average linkage) of genes with standard deviation at top 5% showing a clear separation between A549 cells treated with VS-4718 (2.5 μM) for 12 h (acute) or rendered VS-4718 tolerant as described in (**A**); $n = 3$. (**D**) Analysis of the TRRUST module of Metascape showing that the genes of cluster 5 are identified as transcription factor targets (colored, left) and STRING database analysis showing the possible interaction between the different transcription factors (right). (**E**) Heatmap showing the expression profile of epithelial (KRT8,18) and mesenchymal markers of A549 cells treated as in (**C**). (**F**) Immunoblot for the indicated targets in A549 cells treated as in (**A**). The VS4718-T cells were maintained with 2.5 μM VS-4718. (**G**) Immunoblot for the indicated targets in A549 cells treated as in (**A**) and maintained at 2.5 μM. (**H**) Real-time PCR showing relative mRNA levels of *ERK5* in A549 cells treated as in (**C**); $n = 3$. (**I**) Immunoblot for the indicated targets in A549 parental, VS4718-T and VS4718-T treated with XMD8-92 (10 μM). Heatmaps in (**C, E**) display a relative color scheme across samples that uses the minimum and maximum values in each row to convert the values into a scale ranging from 0 to 1. Graphical data are mean ± SD. Statistical analyses were done using one-way ANOVA; *n*, number of biologically independent samples. Source data are available online for this figure.

analysis of A549 parental and VS4718-T cells evidenced that FAK phosphorylation by CDK5 (pFAK$^{S732}$) was unaltered, while FAK phosphorylation by ERK5 (pFAK$^{S910}$) was increased (Fig. 5G). The increase in ERK5-mediated phosphorylation of FAK was concomitant with a rescue of FAK autophosphorylation (FAK$^{Y397}$) and increased STAT3 signaling, suggesting a compensatory mechanism to sustain FAK activity (Fig. 5G). Moreover, consistent with the increase of FAK phosphorylation by ERK5, the expression of *ERK5* was increased in the VS4718-T group compared to the vehicle control and VS-4718 acutely treated group (Fig. 5H).

Next, to assess whether ERK5 upregulation was responsible for FAK inhibitor tolerance, we treated VS4718-T cancer cells with XMD8-92. Immunoblot analysis evidenced that the inhibition of ERK5 in VS4718-T cells, which reduced ERK5 phosphorylation to a level similar to the parental cells, was sufficient to overcome FAK inhibitor tolerance by repristinating DNA damage and cell death as shown by an increase in γ-H2AX levels and PARP (an apoptotic cell death marker), respectively (Fig. 5I). Taken together, these results suggest that prolonged FAK inhibitor treatment induces drug-tolerant cancer cells via a compensatory ERK5 gain-of-function.

### Pharmacological inhibition of ERK5 improves the anti-tumor response of FAK inhibitors

To assess the relevance of the combined inhibition of FAK and ERK5 in a model of lung adenocarcinoma in vivo, we treated the KP mice 12 weeks after Cre induction with the FAK inhibitor, VS-4718 either alone (FAKi) or in combination with the ERK5 inhibitor, XMD8-92 (FAKi + ERK5i) for 2 weeks (Fig. 6A). Both compounds were well tolerated either alone or in combination (Fig. EV5). As already shown in Fig. 3B,C, the treatment with XMD8-92 alone did not provide any significant anti-tumor effect in lung adenocarcinomas. Conversely, the treatment with the FAKi alone inhibited tumor progression, while the combined treatment with FAKi + ERK5i led to tumor regression as shown by micro-computed tomography (μCT) of the mouse lungs (Fig. 6B). Endpoint measurement of lungs weight and quantification of the tumor size and number from the hematoxylin and eosin (H&E)-stained lungs confirmed the marked tumor regression of the FAKi + ERK5i-treated group compared to the FAKi- or vehicle-treated mice (Fig. 6C,D). Part of the anti-tumor effect of both FAKi- or

FAKi + ERK5i-treated group was due to decreased proliferation as shown by the reduction in Ki-67-positive cancer cells compared to vehicle-treated group (Fig. 6E). Immunoblot assessment of macro-dissected lung tumors at the study endpoint evidenced a consistent increase in cleaved PARP in the FAKi + ERK5i- and partly in the FAKi-treated group, indicating the induction of apoptotic cell death (Fig. 6F). Remarkably, our findings revealed a substantial reduction in FAK and STAT3 activation exclusively within the group treated with both FAKi and ERK5i, in contrast to mice treated solely with the FAK inhibitor or vehicle (Fig. 6F,G). This implies that prolonged FAK inhibitor treatment alone may induce drug resistance and trigger the reactivation of FAK signaling pathways. In aggregate, these results underscore the potential of combining ERK5 and FAK inhibition to overcome potential resistance to FAK inhibitors, thereby enhancing the overall efficacy of anti-tumor responses.

## Discussion

Clinical trials in patients with *KRAS* mutations suggest that, although highly promising, FAK inhibitors do not provide a durable anti-tumor response. Given that FAK is overexpressed in multiple primary solid tumors, and even more in invasive and metastatic cancers (Sulzmaier et al, 2014; Weiner et al, 1993), understanding the mechanisms of FAK inhibitor resistance is of paramount clinical importance. Here, we found that in KRAS-driven lung cancer, FAK activation is synergistically controlled by ERK5 and CDK5. Combined genetic or pharmacological inhibition of ERK5 and CDK5 triggered ROS-induced DNA damage and apoptosis in cancer cells and mouse models of KRAS-driven lung adenocarcinoma, recapitulating FAK pharmacological inhibition. Notably, prolonged treatment of cancer cells with FAK inhibitors induced a drug-tolerant cancer cell state, in which ERK5 was upregulated. Accordingly, ERK5 inhibition was sufficient to break FAK inhibitor tolerant cancer cell state and restore DNA damage-induced cell death.

Our results suggest that ERK5 inhibitors prevent resistance to FAK inhibitors and are expected to provide a good therapeutic strategy for lung cancer patients with *KRAS* mutations. Notably, in NSCLC patients MEK5-ERK5 signalling is often upregulated and correlates with poor patient survival (Sanchez-Fdez et al, 2021),

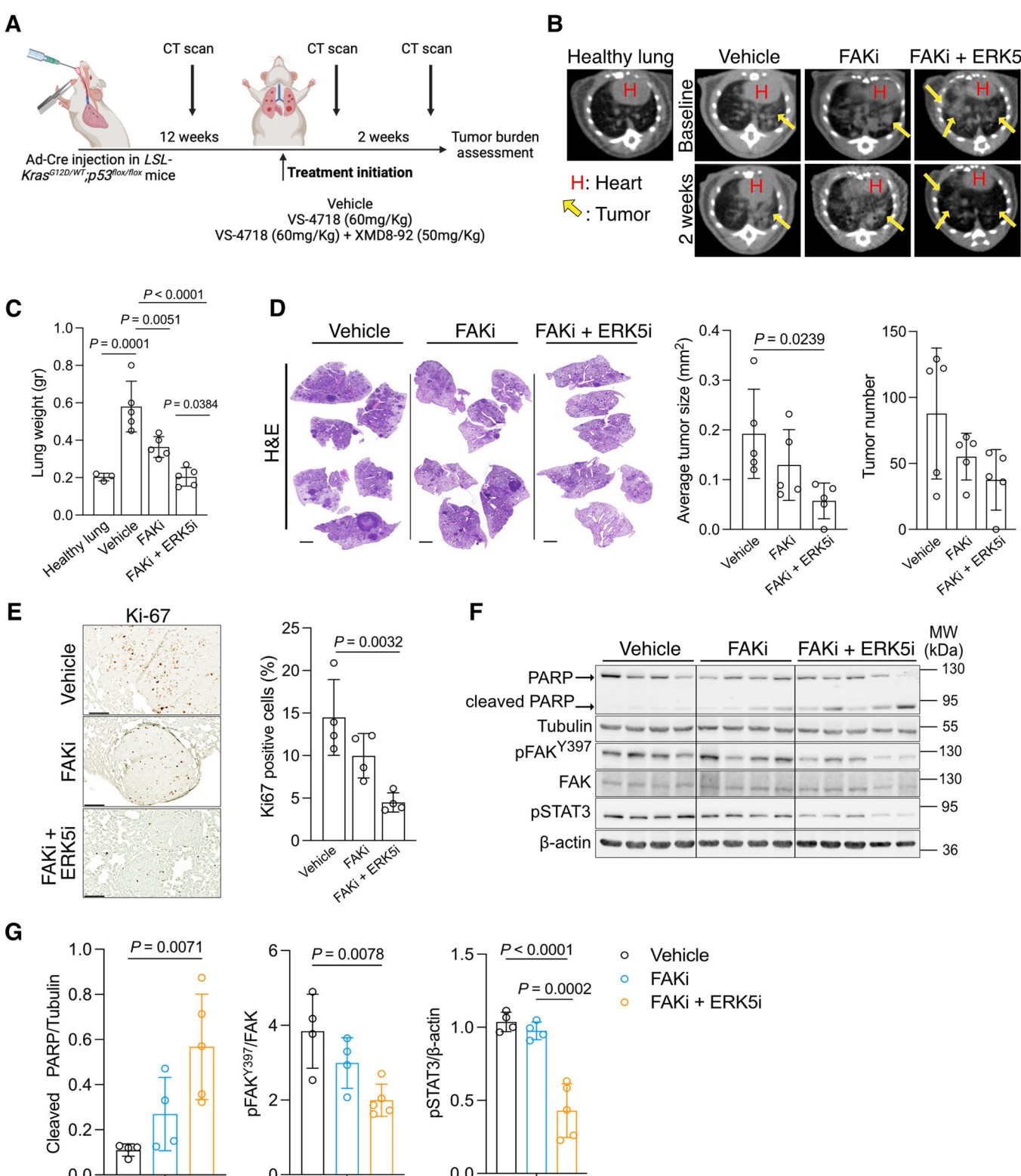

thus these patients are expected to exhibit inherent FAK inhibitor resistance, providing additional rationale for the use of MEK5 or ERK5 inhibitors in the clinic. Furthermore, combination of ERK5 and CDK5 inhibition with XMD8-92 and Seliciclib, synergize in

suppressing FAK and are expected to prevent feedback signaling leading to drug resistance, providing an additional treatment option for lung cancer patients with *KRAS* mutations. Notably, ERK5 inhibitors have been used to sensitize cancer cells in various

**Figure 6. ERK5 inhibition improves the anti-tumor response of FAK inhibitors.**

(A) Scheme of the in vivo experiment workflow. *LSL-Kras*$^{G12D/WT}$*;p53*$^{flox/flox}$ mice were treated once daily with the indicated inhibitors or vehicle starting 12 weeks after Cre induction and continued for 2 weeks. (B) Representative μCT scan images of mouse lungs 2 weeks after treatment initiation (study endpoint). The mice were treated as indicated. A healthy lung image is reported for background reference. Yellow arrows point to tumors. H heart. (C) Lung weight at the study endpoint of *LSL-Kras*$^{G12D/WT}$*;p53*$^{flox/flox}$ mice treated as indicated in (A). The weight of healthy lungs is reported for comparison; n mice/group: 3, 5, 5, 5. (D) Representative hematoxylin & eosin (H&E) staining (left) and quantification of the average tumor size (middle) and tumor number (right) of lung tissue from *LSL-Kras*$^{G12D/WT}$*;p53*$^{flox/flox}$ mice treated as in (A). Scale bars: 2 mm; n mice/group: 5, 5, 5. (E) Representative images of immunohistochemistry against Ki67 (left) and quantification of Ki67-positive cells (right) in lung tissue from *LSL-Kras*$^{G12D/WT}$*;p53*$^{flox/flox}$ mice, treated as in (A). Scale bars: 100 μm; n mice/group: 4, 4, 4. (F, G) Immunoblot analysis (F) and quantification of the indicated targets (G) from macro-dissected lung tumors of *LSL-Kras*$^{G12D/WT}$*;p53*$^{flox/flox}$ mice treated as in (A). In (F) every lane is a different mouse-derived lung lysate, n mice/group: 4, 4, 5. The last treatment with the inhibitors was performed 2–3 h before lung harvesting. Graphical data are mean ± SD. Statistical analyses were done using one-way ANOVA; n: number of mice. Source data are available online for this figure.

combination therapies, including TRAIL-based treatments, further supporting the rationale for developing ERK5 inhibitors for clinical use (Carmell et al, 2021; Espinosa-Gil et al, 2023).

The targeting of CDK5 with Seliciclib is being tested in clinical trials, and it is well tolerated in combination with other drugs such as Gemcitabine or Cisplatin in NSCLC patients (Le Tourneau et al, 2010). However, despite several lead compounds, there are no ERK5 inhibitors in clinical development at the moment (Miller et al, 2023). Therefore, a current limitation is that, in the absence of suitable ERK5 inhibitors, we cannot predict the therapeutic window of these combination therapies in the clinic. Furthermore, given the different off-targets of XMD8-92 (BRD4 and LRRK2) (Miller et al, 2023) and Seliciclib (CDKs), one limitation of our study is the lack of an in vivo system to demonstrate that genetic suppression of ERK5 and/or CDK5 recapitulates the anti-tumor effects observed with the inhibitors and shRNAs in vitro.

Previous evidence suggested that DNA damage induced by ionizing radiation triggers ERK5 upregulation, which protect cells from DNA damage-induced cell death by enhancing the DNA damage response (Jiang et al, 2019). Here, we provided evidence that FAK inhibitors induce apoptosis by triggering the production of high ROS levels and consequently DNA damage, which is counteracted by feedback activation of ERK5. Accordingly, ERK5 suppression repristinates DNA damage and cell death. Future studies are warranted to understand which specific signals trigger ERK5 upregulation during the development of resistance to FAK inhibitors.

After prolonged pharmacologic inhibition of FAK and concomitant to ERK5 signaling activation, we observed STAT3 upregulation. Notably, STAT3 activation has been linked to EMT and therapy resistance (Lee et al, 2014). However, the combination of FAK with ERK5 inhibition not only abolished STAT3 upregulation, but also significantly suppressed it (Fig. 6G). Our results are in line with a previous report in pancreatic cancer, which showed that STAT3 signaling is activated in FAK unresponsive and recurrent tumors (Jiang et al, 2020a). It remains to be investigated whether ERK5, in the context of pancreatic cancer, is also upregulated dampening FAK inhibitor-induced DNA damage and cancer cell death.

Despite the FDA approval of highly specific inhibitors targeting KRAS$^{G12C}$ in the clinic (sotorasib and adagrasib), clinical trial data evidence that not all patients show partial response, owing to the development of drug resistance (Awad et al, 2021). Notably, recent findings indicated that non-genetic acquired resistance to sotorasib involves alterations in focal adhesions (Mohanty et al, 2023).

Therefore, it would be interesting to assess whether addition of FAK inhibitors to KRAS$^{G12C}$ inhibitors would be beneficial in the clinic both to potentiate their efficacy and to prevent the development of non-genetic acquired drug resistance.

## Methods

### Reagents and tools table

| Reagent/resource | Reference or source | Identifier or catalog number |
|---|---|---|
| **Experimental models** | | |
| Human: A549 cell line | ATCC | CCL-185 |
| Human: HEK293T cell line | ATCC | CRL-11268 |
| Human: Phoenix-Ampho cell line | ATCC | CRL-3213 |
| Human: A427 cell line | Prof. John Minna (UTSW Medical Center, Dallas, USA) | ATCC # HTB-53 |
| Human: H460 cell line | Prof. John Minna (UTSW Medical Center, Dallas, USA) | ATCC # HTB-177 |
| Human: HBEC3KT cell line | Prof. John Minna (UTSW Medical Center, Dallas, USA) | ATCC # CRL-4051 |
| *B6.129SS4-kras*$^{tm4Tyj}$*/J* | The Jackson Laboratory | 008179 |
| *B6.129P2-Trp53*$^{tm1Bm}$*/J* | The Jackson Laboratory | 008462 |
| **Recombinant DNA** | | |
| pCMV-VSV-G | Stewart et al, 2003 | Addgene Plasmid #8454; RRID:Addgene_8454 |
| pCMV-dR8.2 dvpr | Stewart et al, 2003 | Addgene Plasmid #8455; RRID:Addgene_8455 |
| pSpCas9(BB)-2A-GFP (PX458) | Ran et al, 2013 | Addgene Plasmid #48138; RRID:Addgene_48138 |
| pLVUHshp53 | Szulc et al, 2006 | Addgene plasmid #11653; RRID:Addgene_11653 |
| pBABE-Zeo | Addgene | #1766; RRID:Addgene_1766 |
| pBABE-KRASG12D-Zeo | This study | NA |
| pLKO.1 hygro | Addgene | #24150; RRID:Addgene_24150 |
| pLKO.1 hygro-shRNA ERK5 | This study | NA |
| Tet-PLKO-puro | Wiederschain et al, 2009 | Addgene Plasmid #21915; RRID:Addgene_21915 |
| Tet-pLKO-puro-shRNA CDK5 | This study | NA |
| pWZL-Hygro | Addgene | #18750; RRID:Addgene_18750 |
| pWZL-hygro-FAK$^{WT}$ | This study | Deposited at Addgene, plasmid #216540 |
| pWZL-hygro-FAK$^{S732A}$ | This study | Deposited at Addgene, plasmid #216541 |
| pWZL-hygro-FAK$^{S910A}$ | This study | Deposited at Addgene, plasmid #216542 |

| Reagent/resource | Reference or source | Identifier or catalog number |
|---|---|---|
| pWZL-hygro-FAK$^{R177/178A}$ | This study | Deposited at Addgene, plasmid #216543 |
| pWZL-hygro-FAK$^{P712/713A}$ | This study | Deposited at Addgene, plasmid #216544 |
| pWZL-hygro-FAK$^{S722A}$ | This study | Deposited at Addgene, plasmid #216545 |
| pWZL-hygro-FAK$^{Y861F}$ | This study | Deposited at Addgene, plasmid #216546 |
| pWZL-hygro-FAK$^{Y925F}$ | This study | Deposited at Addgene, plasmid #216547 |
| pWZL-hygro-FAK$^{Y397F}$ | This study | Deposited at Addgene, plasmid #216677 |
| **Antibodies** | | |
| Rabbit Monoclonal Anti-Ki67 clone SP6 (IHC: 1/300) | Thermo Scientific | Cat# RM-9106, RRID:AB_2341197 |
| Mouse monoclonal Anti-FAK (D-1) (WB: 1/1000) | Santa Cruz Biotechnology | Cat# sc-271126, RRID:AB_10614323 |
| Rabbit monoclonal Anti-RAS (27H5) (WB: 1/1000) | Cell Signaling Technology | Cat# 3339, RRID:AB_2269641 |
| Mouse monoclonal Anti-β-actin (AC-74) (WB: 1/2000) | Sigma-Aldrich | Cat# a-5316, RRID:AB_476743 |
| Mouse monoclonal α-tubulin (WB: 1/2000) | Sigma-Aldrich | Cat# T6199, RRID:AB_477583 |
| Rabbit monoclonal anti-Phospho-STAT3 D8C2Z (Ser727) (WB: 1/1000) | Cell Signaling Technology | Cat# 94994, RRID:AB_2800239 |
| Rabbit monoclonal Anti-Cyclin D1 (WB: 1/1000) | Cell Signaling Technology | Cat# 2978, RRID:AB_2259616 |
| Rabbit polyclonal Anti-Phospho-FAK (Tyr397) (WB: 1/1000) | Cell Signaling Technology | Cat# 8556, RRID:AB_10891442 |
| Rabbit polyclonal Anti-Phospho-FAK (Ser910) (WB: 1/1000) | Invitrogen | Cat# 44-596G, RRID:AB_2533689 |
| Rabbit monoclonal Anti-Vimentin (D21H3) (WB: 1/1000) | Cell Signaling Technology | Cat# 5741, RRID: AB_10695459 |
| Rabbit monoclonal Anti-E-Cadherin (24E10) (WB: 1/1000) | Cell Signaling Technology | Cat# 3195, RRID: AB_2291471 |
| Rabbit monoclonal Anti-PARP (46D11) (WB: 1/1000) | Cell Signaling Technology | Cat# 9532, RRID:AB_659884 |
| Rabbit polyclonal Anti-Phospho-FAK (Ser732) (WB: 1/1000) | Invitrogen | Cat# 44-590G, RRID:AB_2533688 |
| Rabbit polyclonal Anti-Phospho-H2AX (Ser139) (WB:1/1000) | Cell Signaling Technology | Cat# 2577, RRID:AB_2118010 |
| Mouse monoclonal Anti-p21 (WB: 1/1000) | BD Pharmigen | Cat# 556430, RRID:AB_396414 |
| Mouse monoclonal Anti-CDK5 (J-3) (WB: 1/1000) | Santa Cruz Biotechnology | Cat# sc-6247, RRID:AB_627241 |
| Rabbit Anti-phospho-ERK5 (Thr218/Tyr220) (WB: 1/1000) | Cell Signaling Technology | Cat# 3371, RRID:AB_2140424 |
| Mouse monoclonal Anti-ERK5 (C-7) (WB: 1/1000) | Santa Cruz Biotechnology | Cat# sc-398015 |
| Mouse monoclonal Anti-p53 (DO-1) (WB: 1/1000) | Santa Cruz Biotechnology | Cat# sc-126, RRID:AB_628082 |
| ImmPRESS® HRP Anti-Rabbit IgG (Peroxidase) | Vector labs | Cat# MP-7401, RRID:AB_2336529 |
| Goat Anti-mouse Red IRDye 800 CW (WB: 1/10000) | LI-COR | Cat# 926-32210, RRID:AB_621842 |
| Goat Anti-mouse Green IRDye 680 RD (WB: 1/10000) | LI-COR | Cat# 926-68070, RRID:AB_10956588 |
| Goat Anti-rabbit Red IRDye 800 CW (WB: 1/10000) | LI-COR | Cat# 926-32211, RRID:AB_621843 |
| Goat Anti-rabbit Green IRDye 680 RD (WB: 1/10000) | LI-COR | Cat# 926-68071, RRID:AB_10956166 |
| Amersham ECL Mouse IgG, HRP-linked whole Ab | GE Healthcare Life Sciences | Cat# NA931, RRID:AB_772210 |
| Amersham ECL Rabbit IgG, HRP-linked whole Ab | GE Healthcare Life Sciences | Cat#NA934, RRID:AB_772206 |
| **Oligonucleotides and other sequence-based reagents** | | |
| shRNA target sequence against human ERK5 (MAPK7): GTTCATCTCAG ACCCACCTTT | Sigma-Aldrich | TRCN0000197264 |
| shRNA target sequence against human CDK5: CCTGAGATTG TAAAGTCATTC | Sigma-Aldrich | TRCN0000194974 |
| shRNA target sequence against human TP53: AGTAGATTACC ACTGGAGTCTT | Addgene | Addgene plasmid #11653; RRID:Addgene_11653 |
| sgRNA FAK #1: 5'-AGAGCAAAAG ATTTGTACAC-3' | Sigma-Aldrich | NA |
| sgRNA FAK #2: 5'-ATGTGGGAGATACTG ATGCA-3' | Sigma-Aldrich | NA |
| RT - PCR oligo for Human ERK5 (MAPK7) FW: 5'-AGCAGGTGGCCATC AAGAAG-3' | Sigma-Aldrich | NA |
| RT - PCR oligo for Human ERK5 (MAPK7) RV: 5'-CAGGACCACG TAGACAGATTT-3' | Sigma-Aldrich | NA |
| RT- PCR oligo for Human β-ACTIN FW: AGAGCTACGA GCTGCCTGAC | Sigma-Aldrich | NA |
| RT-PCR oligo for Human β-ACTIN RV: AGCACTGTGTT GGCGTACAG | Sigma-Aldrich | NA |
| **Chemicals, enzymes and other reagents** | | |
| TransIT®-293 Transfection Reagent | Mirus Bio | Cat# MIR2705 |
| Immobilon Forte Western HRP Substrate | Millipore | Cat# WBUF0500 |
| Methyl Green | Vector Labs | Cat# H-3402 |
| Annexin V-Atto 633 | In house | N/A |
| Propidium Iodide (PI) | Invitrogen | Cat# P3566 |
| Puromycin | Gibco | Cat# A11138-03 |
| Hygromycin B | Invitrogen | Cat# 10687010 |
| DAB+ solution | Dako | Cat# K3467 |
| Doxycycline (1 μg/mL) | Fisher BioReagents | Cat# BP2653 |
| XMD8-92 for in vitro | Tocris #4132 | Cat# 4132 |
| Seliciclib for in vitro | AdipoGen | Cat# AG-CR1-0006-M001 |
| PF-562271 | Sigma | Cat# PZ0387 |
| VS-4718 (for in vivo 60 mg/Kg per day) | MedChemExpress | Cat# HY-13917 |
| MnTMPyP (SOD mimetic) | Sigma-Aldrich | Cat# 475872 |
| Dihydrorhodamine 123 (DHR) | Sigma-Aldrich | Cat# D1054 |
| Zeocin | InvivoGen | Cat# ant-zn-1 |
| XMD8-92 for in vivo (50 mg/Kg) | MedChemExpress | Cat# HY-14443 |
| Seliciclib for in vivo (50 mg/Kg) | MedChemExpress | Cat# HY-30237 |
| TransIT®-LT1 Transfection Reagent | Mirus Bio | Cat# MIR2300 |
| PureLink™ RNA Mini Kit | Invitrogen | Cat# 12183018A |
| PureLink™ DNase Kit | Invitrogen | Cat# 12185-010 |
| cDNA Synthesis Kit | Thermo Scientific | Cat# K1622 |

| Reagent/resource | Reference or source | Identifier or catalog number |
|---|---|---|
| KAPA HotStart Mouse Genotyping Kit | Kapa Biosystems | Cat# KK7352 |
| FastSybr© green | Thermo Scientific | Cat# 4367659 |
| The DeadEnd™ Colorimetric TUNEL System | Promega | Cat# G7130 |
| **Software** | | |
| QuPath v.0.1.2 | https://qupath.github.io/ | |
| FlowJo V10 | https://www.flowjo.com/ | |
| GraphPad Prism v.7 | https://www.graphpad.com/scientific-software/prism/ | |
| TIDE: Tracking of Indels by Decomposition | http://shinyapps.datacurators.nl/tide/ | |
| Compusyn | http://www.combosyn.com | |
| ImageJ | https://imagej.nih.gov/ij/ | |
| Biorender | https://app.biorender.com | |
| fastqc v.0.11.9 | http://www.bioinformatics.babraham.ac.uk/projects/fastqc | |
| RSeQC v.4.0.0 | https://rseqc.sourceforge.net | |
| HiSat2 v.2.2.1 | http://daehwankimlab.github.io/hisat2/ | |
| FeatureCounts v.2.0.1 | http://subread.sourceforge.net/ | |
| Bioconductor package DESeq2 v1.38.1 | https://bioconductor.org/packages/release/bioc/html/DESeq2.html | |
| ClusterProfiler v4.6.0 | https://bioconductor.org/packages/release/bioc/html/clusterProfiler.html | |
| Shiny application v1.6.0 | https://shiny.posit.co/ | |
| R version 4.2.1 | https://www.r-project.org/ | |
| Metascape | https://metascape.org/gp/index.html#/main/step1 | |
| TRRUST | https://www.grnpedia.org/trrust/Network_search_form.php | |
| **Other** | | |

## Plasmids and cloning

The plasmid pBABE-zeo (Addgene Plasmid #1766; https://www.addgene.org/1766/; RRID:Addgene_1766) was a gift from Hartmut Land & Jay Morgenstern & Bob Weinberg (Morgenstern and Land, 1990). The pWZL-Hygro (Addgene Plasmid #18750; https://www.addgene.org/18750/; RRID:Addgene_18750) was a gift from the Scott Lowe. pSpCas9(BB)-2A-GFP (PX458) (Addgene Plasmid #48138; https://www.addgene.org/48138/; RRID:Addgene_48138) was a gift from Feng Zhang (Ran et al, 2013). Tet-pLKO-puro (Addgene Plasmid #21915, https://www.addgene.org/

21915/; RRID:Addgene_21915) was a gift from Prof. Dmitri Wiederschain (Wiederschain et al, 2009). pLKO.1 hygro (Addgene plasmid #24150; https://www.addgene.org/24150/; RRID:Addgene_24150), pCMV-VSV-G (Addgene plasmid #8454; https://www.addgene.org/8454/; RRID:Addgene_8454), and pCMV-dR8.2 dvpr (Addgene plasmid #8455; https://www.addgene.org/8455/; RRID:Addgene_8455) were a gift from Prof. Bob Weinberg (Stewart et al, 2003). The pLVUHshp53 was a gift from Patrick Aebischer & Didier Trono (Addgene plasmid # 11653; http://n2t.net/addgene:11653; RRID:Addgene_11653) (Szulc et al, 2006). KRAS$^{G12D}$ was amplified from pDONR223_KRAS_p.G12D (Addgene, Plasmid: #81651) and ligated into pBABE-zeo (Addgene, Plasmid: #1766) using the BamHI and SalI restriction sites. The *PTK2* point mutants sequences were designed and ordered from GenScript, amplified and cloned into pWZL-Hygro using the sequence- and ligation-independent cloning (SLIC) method (Li and Elledge, 2012). The complete list of plasmids used are reported in the Reagents Table.

## Cell lines

HBEC3-KT, A427 and H460 human cell lines were kindly provided by Dr. John Minna (UT Southwestern Medical Center). A549 (CCL-185), HEK293T (CRL-11268) and Phoenix-Ampho cells (CRL-3213) cell lines were purchased from ATCC. The HBEC3-KT cells were cultured in KSFM (Gibco) supplemented with EGF, Bovine Pituitary extract (Gibco) (Ramirez et al, 2004). The human NSCLC cell lines A549, A427 and H460 were cultured in RPMI-1640 (Gibco) supplied with 10% FBS (Thermo Fisher), 100 I.U./mL penicillin and 100 μg/mL streptomycin (Gibco). All cell lines were DNA fingerprinted for provenance, screened for mycoplasma and cultured in an incubator at 37 °C and 5% $CO_2$.

## shRNAs, virus production and transduction

The HEK293T cells were transfected with pCMV-VSV-G (VSV-G protein), pCMV-dR8.2 (lentivirus packaging vector) and lentiviral constructs Tet-pLKO-puro, Tet-pLKO-puro-shRNA CDK5, pLKO.1 hygro and pLKO.1 hygro-shRNA ERK5 to generate lentiviruses. Cancer cell lines were then infected and selected with 150 μg/ml Hygromycin B (Thermo Scientific) and 2 μg/mL puromycin (Thermo Scientific). After selection, the shRNA in the Tet-pLKO-puro was induced with 1 μg/mL doxycycline. The Phoenix-Ampho cells were used for production of the pWZL-Hygro and pBABE-zeo retroviruses. The transfection of the different constructs was done using the TransIT®-293 Transfection Reagent (Mirus; MIR 2705), according to the manufacturer's instructions. The list of shRNA sequences used are reported in the Reagents Table.

## Immunohistochemistry

Immunohistochemistry was performed on paraffin-embedded tissue and sections were 5μm thick. Tumor burden was assessed using QuPath v.0.1.2 (Bankhead et al, 2017) which quantifies the area occupied by tumors compared to unaffected tissue. Sections were deparaffinized, rehydrated and exposed to antigen retrieval by boiling for 10 min in Sodium Citrate buffer (pH 6). After this step, the sections were pretreated for 30 min with 3% hydrogen peroxide

(Sigma-Aldrich, 216763) in PBS, washed twice with 0.1 M Tris-Buffered Saline (TBS), blocked for 1 h in 2% bovine serum albumin (BSA) in TBS containing 0.1% Polysorbate 20 (TBS-T), followed by 10 min in 2.5% normal horse serum (Vector, S-2012) and finally incubated with primary antibodies in blocking solution. The following day, sections were washed in TBS-T, incubated with secondary antibody (Vector, MP-7401) for 10 min and the staining was revealed with DAB+ solution (Dako, K3467). Tissue sections were then counterstained with Methyl Green (Vector Labs, H-3402), followed by dehydration and mounting. The list of antibodies used in IHC are reported in the Reagents Table.

## Immunoblotting

Cells were lysed in RIPA buffer (50 mM Tris-HCl pH 8.0, 150 mM NaCl, 1.0% NP-40, 0.5% sodium deoxycholate, 0.1% SDS) or NP40 buffer (50 mM Tris-HCl pH 8.0, 150 mM NaCl, 1% NP-40) with complete EDTA-free protease inhibitors (Roche) and 1 mM PMSF. Samples were resolved by SDS-PAGE in Bio-Rad blotting chamber, transferred to nitrocellulose membrane using a semi-dry chamber (Bio-Rad) and blocked in 5% BSA in PBS containing 0.1% Tween. Membranes were then incubated overnight at 4 °C with primary antibody diluted in 5% BSA in PBS containing 0.1% Tween. The immunoblot development was done using LI-COR fluorescence-chemiluminescence detector. The complete list of primary and secondary antibodies used is included in the Reagents Table.

## Cell proliferation, clonogenic and migration assays

For the cell proliferation assay cells were plated at low confluency either in 24-well plates (8000 cells/well for NSCLC cancer cells and 3500 cells/well for HBEC3-KT) or in 48-well plates (3500 cells/well for NSCLC cancer cells) in triplicates and let proliferate for 2–4 days as indicated in the related figure legends. Relative cell number was measured by crystal violet (Sigma-Aldrich) staining (0.1% in 20% methanol) of adherent cells after 10 min fixation in 4% paraformaldehyde (Sigma-Aldrich). After washing twice and air-drying, stained cells were de-colored with 10% acetic acid and OD600 was measured with a spectrophotometer. Best-fit curves were generated in GraphPad Prism [(log (inhibitor) versus response (-variable slope three parameters)].

For the clonogenic assays, 700–1000 cells/well were plated in 6-well plates. Quantification of the colonies (colony ≥50 cells) was performed with ImageJ using the 'cell counter' plugin function.

The migration assay was performed using a Transwell system (8-μm pore size Pore Polycarbonate Membrane Insert; Catalog. 3422, Corning). Cells were starved overnight the day before the assay. In total, $4 \times 10^4$ cells were seeded onto the Transwell chamber in Opti-MEM™ I Reduced Serum Medium. RPMI supplemented with 10% FBS was added in the bottom chamber to act as a chemoattractant. The cells were allowed to migrate for 12 h upon treatment with DMSO (control), XMD8-92, and Seliciclib either alone or in combination. The cells that did not migrate were scraped off using wetted cotton swabs, while the migrated cells were fixed with 4% paraformaldehyde followed by DAPI staining and microscope analysis.

## Cell death and ROS measurements

For the Annexin V/PI cell death assay, $3 \times 10^4$ cells were plated in a 12-well plate and treated with pharmacological inhibitors (for 72 h

for A549 and 24 h for A427) or transduced as described in the figure legends. On the day of the assay, cells were washed with staining buffer (150 mM NaCl, 4 mM KCl, 2.5 mM CaCl$_2$, 1 mM MgSO$_4$, 15 mM HEPES pH 7.2, 2% FBS, and 10 mM NaN3) and stained with Atto633-conjugated Annexin V for 20 min in the dark, on ice. Cells were then washed with staining buffer and resuspended in 200 μL Propidium Iodide (PI) at a final concentration of 4 μg/mL (A427), 20 μg/mL (A549) or 40 μg/mL (HBEC3). In Fig. 1, all gated populations were included in the analysis, whereas in the rest of the figures only Annexin V+ and Annexin V + /PI+ populations were included in the analysis. For the cell death measurement on mouse tissue, we used the DeadEnd™ Colorimetric TUNEL System according to the manufacturer's instructions (Promega, G7130).

To measure intracellular ROS levels, $1 \times 10^5$ cells were resuspended in 1 μM Dihydrorhodamine 123 (DHR) (Sigma-Aldrich, D1054) in PBS and incubated for 30 min at 37 °C. To quench ROS, cells were treated with 25 μM MnTMPyP (Sigma, # 475872) for 18 h or 24 h prior to staining.

## CRISPR/Cas9-mediated genome editing

To knockout FAK, the HBEC3-KT cells were transfected with pSpCas9(BB)-2A-GFP (PX458) vector (Addgene plasmid #48138) using the TransIT-LT1 Transfection reagent (Catalog. MIR 2305) in which two validated sgRNAs (5'-AGAGCAAAAGATTTGTA-CAC-3'; 5'-ATGTGGGAGATACTGATGCA-3') were cloned using the BbsI restriction site between the U6 promoter and the gRNA scaffold, following the CRISPR-Cas9 system protocol (Ran et al, 2013). Seventy-two hours post-transfection, single cells were isolated by FACS sorting of GFP-positive cells on a FACS Aria instrument and plated into 96-well plates. After expansion of the single cell clones, genomic DNA was isolated and a PCR of the sgRNA target region was performed. Gene editing was confirmed by Sanger sequencing/tracking of indels by decomposition (TIDE) analysis (Brinkman et al, 2014) and immunoblotting.

## Animal studies

Mice were maintained under a temperature of 21 °C + /− 2 °C, humidity 50% +/− 10% with a standard 12 h light/dark cycle and were fed *ad libitum*. The KAPA HotStart Mouse Genotyping Kit (Kapa Biosystems, KK7352) and KAPA2G Fast HotStart Genotyping Mix (Kapa Biosystems, KK5621) were used to perform the genotyping, according to the manufacturer's instructions. Mixed background *LSL-Kras*$^{G12D/WT}$;*p53*$^{flox/flox}$ mice were generated by crossing stock *B6.129SS4-kras*$^{tm4Tyj}$/J (from JaxLab, Stock number 008179) (Jackson et al, 2001), with *B6.129P2-Trp53*$^{tm1Brn}$/J (from JaxLab, Stock number 008462) (Marino et al, 2000) mice. Mixed sex littermates were used for the experiments. All the mouse experiments were randomized. Animals were first genotyped and then randomly assigned to each group after balancing of age. Blinding in mouse studies was not possible due to tagging of the mice and access to identification codes by the investigators.

For intratracheal injections, $2.5 \times 10^7$ infectious particles of VVC-U of Iowa-5 Ad5CMVCre (Viral Vector Core, University of Iowa) were delivered to the mice at 8 weeks of age. For the experiment in Fig. 3, 10 weeks after Cre induction, mice were intraperitoneally injected once daily with vehicle (10% DMSO, 40% PEG 300; 5% Tween 80 and

45% Saline), XMD8-92 (HY-14443, MedChemExpress) and/or Seliciclib (MedChemExpress, HY-30237) at a dosage of 50 mg/kg (in 100 µl volume) for a period of 2 weeks. XMD8-92 and Seliciclib were resuspended in DMSO, aliquoted and stored at -20C. For the experiment in Fig. 6, 12 weeks after Cre induction, mice were intraperitoneally injected once daily with vehicle (20% DMSO, 40% PEG 300 and 40% Saline) or XMD8-92 at a dosage of 50 mg/kg and/or VS-4718 delivered via oral gavage (HY-13917, MedChemExpress) at a dosage of 60 mg/kg (in 100 µl volume) for a period of 2 weeks. The preparation for the drug delivery to the mice was done the day of the treatment by adding first the PEG 300, followed by Tween 80 and saline. To follow tumor development during treatment with µCT scan, 12 weeks after Cre instillation, mice were scanned once a week using an X-RAD SmART Precision X-Ray Imaging System (Precision X-Ray, North Bradford, CT, USA). Briefly, mice were anesthetized with 2–2.5% isoflurane during the entire procedure and images acquired after photon filtering using a 2 mm AI filter for computed tomography. After, the raw DICOM data was analyzed with 3D Slicer version 5.6.1. (Velazquez et al, 2013). The study is compliant with all relevant ethical regulations regarding animal research. Experimental procedures were approved by the cantonal veterinary commission and animal welfare officer from the Veterinaerdienst de Kantons Bern (animal protocol: BE133/2020).

## Bulk RNA sequencing and analysis

Prior to RNA sequencing, A549 cells were treated overnight with vehicle (DMSO), XMD8-92 (10 µM), Seliciclib (10 µM), alone or in combination (Fig. 4) or were treated either acutely with VS-4718 (12 h) or rendered resistant by treating them with increasing doses of VS-4718 overtime to select for drug-tolerant subpopulations (VS4718-T) (Fig. 5). For the RNA sequencing, total RNA was extracted using a PureLink™ RNA Kit (Invitrogen, 12183018A), including an on-column DNase treatment (Invitrogen, 12185-010) according to the manufacturer's instructions. The quantity and quality of the purified total RNA was assessed using a Thermo Fisher Scientific Qubit 4.0 fluorometer with the Qubit RNA BR Assay Kit (Thermo Fisher Scientific, Q10211) and an Advanced Analytical Fragment Analyzer System using a Fragment Analyzer RNA Kit (Agilent, DNF-471), respectively. Sequencing libraries were made with 1000 ng input RNA using an Illumina TruSeq Stranded mRNA Library Prep kit (Illumina, 20020595) in combination with TruSeq RNA UD Indexes (Illumina, 20022371) according to Illumina's guidelines. The cDNA libraries were evaluated using a Thermo Fisher Scientific Qubit 4.0 fluorometer with the Qubit dsDNA HS Assay Kit (Thermo Fisher Scientific, Q32854 and an Agilent Fragment Analyzer (Agilent) with a HS NGS Fragment Kit (Agilent, DNF-474), respectively. Equimolar-pooled cDNA libraries were sequenced paired-end using a shared Illumina NovaSeq 6000 SP Reagent Kit (100 cycles; Illumina, 20028401) on an Illumina NovaSeq 6000 instrument. An average of 37 million reads were generated/library. The quality of the sequencing run was assessed using Illumina Sequencing Analysis Viewer (Illumina version 2.4.7) and all base call files were demultiplexed and converted into FASTQ files using Illumina bcl2fastq conversion software v2.20. The quality control assessments, generation of libraries and sequencing was conducted by the Next Generation Sequencing Platform of the University of Bern.

The quality of the RNA-seq data was assessed using fastqc v.0.11.9 (Andrews, 2022) and RSeQC v.4.0.0 (Wang et al, 2012). The reads were mapped to the reference genome using HiSat2 v.2.2.1 (Kim et al, 2015). FeatureCounts v.2.0.1 (Liao et al, 2014) was used to count the number of reads overlapping with each gene as specified in the genome annotation (Homo_sapiens. GRCh38.104). The Bioconductor package DESeq2 v1.38.1 (Love et al, 2014) was used to test for differential gene expression between the experimental groups. Gene set enrichment analysis (GSEA) (Subramanian et al, 2005) was run in ClusterProfiler v4.6.0 (Wu et al, 2021) using genesets from KEGG (Kanehisa et al, 2022). An interactive Shiny application v1.6.0 (https://shiny.posit.co/) was set up to facilitate the exploration and visualisation of the RNA-seq results. All analyses were run in R version 4.2.1 https://www.R-project.org/. For Fig. 5D, the gene annotation and analysis resource, metascape was used to generate the functional categories associated with the different clusters.

## Reverse transcription and qPCR

RNA was extracted using the RNAeasy kit (Qiagen, 74104) and cDNA was synthesized with the RevertAid First Strand cDNA Synthesis Kit (Thermo Scientific, K1622). QPCR was performed in 96-well plates (TreffLab) with FastSybr green (Thermo Scientific, 4367659). The normalization was performed with the ΔΔCT method.

## Statistics and reproducibility

All data sets were organized and analyzed in Microsoft excel version 16.73 and GraphPad Prism version 7.0.0 (GraphPad Software, San Diego, California USA, www.graphpad.com). All data presented are expressed as mean ± SD (unless indicated in the figure legends) of 3 or more biologically independent replicates/group (the specific number is indicated in the related figure legends). All graphical in vitro data report 1 representative experiment from experiments performed independently at least three times (except for the RNA sequencing that was performed once) with similar results. The significance of the results was determined employing two-tailed unpaired Student's $t$ test (when comparing two groups) or one-way ANOVA Tukey's correction for multiple comparisons (when more than two groups were compared) and significance is indicated directly in the figure panels and legends. For the cell proliferation data only the figure-relevant statistical differences are represented. No outliers were found in any dataset and no animals or data were excluded from statistical analysis.

## Quantitative analysis of drug synergy

Drug synergism was calculated using CompuSyn software (version 1.0) (http://www.combosyn.com), which is based on the median-effect principle (Chou) and the combination index-isobologram theorem (Chou–Talalay) (Chou, 2010). CompuSyn creates combination index (CI) values, where CI < 0.75 indicates synergism, CI = 0.75–1.25 additive effects, and CI > 1.25 antagonism.

# Data availability

All data are available in the main text or the supplementary materials. The bulk RNA sequencing data have been deposited in the NCBI's Gene Expression Omnibus (Edgar et al, 2002) and are accessible through GEO Series accession number GSE255628 and

## The paper explained

### Problem

Lung tumors that are driven by the oncogene KRAS are the most aggressive and refractory to therapy. Targeted therapies against KRAS-driven tumors invariably fail due to therapy resistance. Focal adhesion kinase (FAK) is a non-receptor tyrosine kinase that is required to sustain KRAS-driven tumors. However, clinical trials evidenced that the efficacy of FAK inhibitors in producing long-term anti-tumor responses has been limited in the clinic, suggesting the development of drug resistance. The molecular basis of this resistance is a matter of ongoing investigation.

### Results

We found that the combined inhibition of ERK5 and CDK5 synergistically suppressed FAK function, decreased proliferation and caused DNA damage, which resulted in cell death in human cancer cell lines and mouse models of lung cancer. We found that cancer cells resistant to FAK inhibitors evidence enhanced ERK5-FAK signaling dampening DNA damage. Notably, ERK5 inhibition was sufficient to prevent resistance of cancer cells to FAK inhibitors.

### Impact

This study adds new insights into the molecular mechanisms that govern the resistance of lung tumors to FAK inhibitors in the clinic. We propose ERK5 inhibition as a potential co-targeting strategy to counteract FAK inhibitor resistance in lung cancer patients with KRAS mutations. Notably, ERK5 has been already found upregulated in KRAS-lung cancer, additionally supporting the need to target it.

GSE255643. The human PTK2 point mutant constructs have been deposited at addgene. The plasmid IDs are reported in the Reagents Table. Any other material requests should be addressed to Georgia Konstantinidou georgia.konstantinidou@unibe.ch.

The source data of this paper are collected in the following database record: biostudies:S-SCDT-10_1038-S44321-024-00138-7.

# Peer review information

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

## Acknowledgements

We thank the Interfaculty Bioinformatics Unit (IBU) of the University of Bern for providing computational infrastructure and support with bioinformatic analyses, and Ioanna Nikdima for help with drug delivery in mice. The study has been supported by Swiss National Foundation (SNF) Professorship #PP00P3_163929, #PP00P3_194810 and SNF project grant #310030_212418 to GK.

## Author contributions

**Chiara Pozzato**: Data curation; Formal analysis; Methodology; Writing—original draft; Writing—review and editing. **Gonçalo Outeiro-Pinho**: Data curation; Formal analysis; Validation; Methodology; Writing—review and editing. **Mirco Galiè**: Resources; Data curation; Software; Formal analysis; Visualization; Methodology. **Giorgio Ramadori**: Investigation; Writing—review and editing. **Georgia Konstantinidou**: Conceptualization; Supervision; Funding acquisition; Investigation; Project administration; Writing—review and editing.

Source data underlying figure panels in this paper may have individual authorship assigned. Where available, figure panel/source data authorship is listed in the following database record: biostudies:S-SCDT-10_1038-S44321-024-00138-7.

## Disclosure and competing interests statement

The authors declare no competing interests.

# Expanded View Figures

**Figure EV1. Functional characterization of the different human FAK phospho-mutants.**

(A) Immunoblot analysis of the indicated targets of HBEC3-FAK KO cell line transduced with a lentiviral plasmid carrying either a control shRNA (scramble) or a shRNA against human p53. (B) Immunoblot analysis of the indicated targets in parental and FAK knockout HBEC3 cell line transduced with either an empty vector or a plasmid encoding mutant KRAS (left) and relative cell number of the indicated groups calculated at day 4 (right); $n = 3$. MutKRAS: mutant KRAS; ns: not significant. (C) Relative cell number of HBEC3-FAK KO cell line transduced either with empty vector (pWZL-Hygro) or with the FAK phospho-mutant Y397F (pWZL-Hygro-FAK Y397F) in the absence or presence of mutKRAS (pBABE-zeo or pBABE-zeo-KRAS $^{G12D}$). MutKRAS: mutant KRAS; $n = 3$. Note that the Empty vector and Empty vector + mutKRAS group plots are the same as in main Fig. 1E because these experiments were performed at the same time for direct comparison. (D) Relative quantification of colony forming capacity of HBEC3-FAK KO cells previously transduced as indicated; $n = 3$. (E) Immunoblot analysis of the indicated targets in HBEC3-FAK KO cells transduced either with empty vector (pWZL-Hygro) or with the FAK phospho-mutant Y397F (pWZL-Hygro-FAK Y397F). (F) Immunoblot analysis of the indicated targets in HBEC3-FAK KO transduced either with empty vector (pWZL-Hygro) or wild-type FAK (pWZL Hygro-FAK WT) or with the indicated human FAK phospho-mutants. (G–J) Relative cell number of HBEC3-FAK KO cell line transduced either with wild-type FAK (pWZL Hygro-FAK WT) or the indicated human FAK phospho-mutants in the absence or presence of mutKRAS$^{G12D}$ (pBABE-zeo or pBABE-zeo-KRAS $^{G12D}$). MutKRAS: mutant KRAS; $n = 3$. Graphical data are mean ± SD. Statistical analyses were done using two-tailed unpaired Student's $t$ test or one-way ANOVA; $n$, number of biologically independent samples.

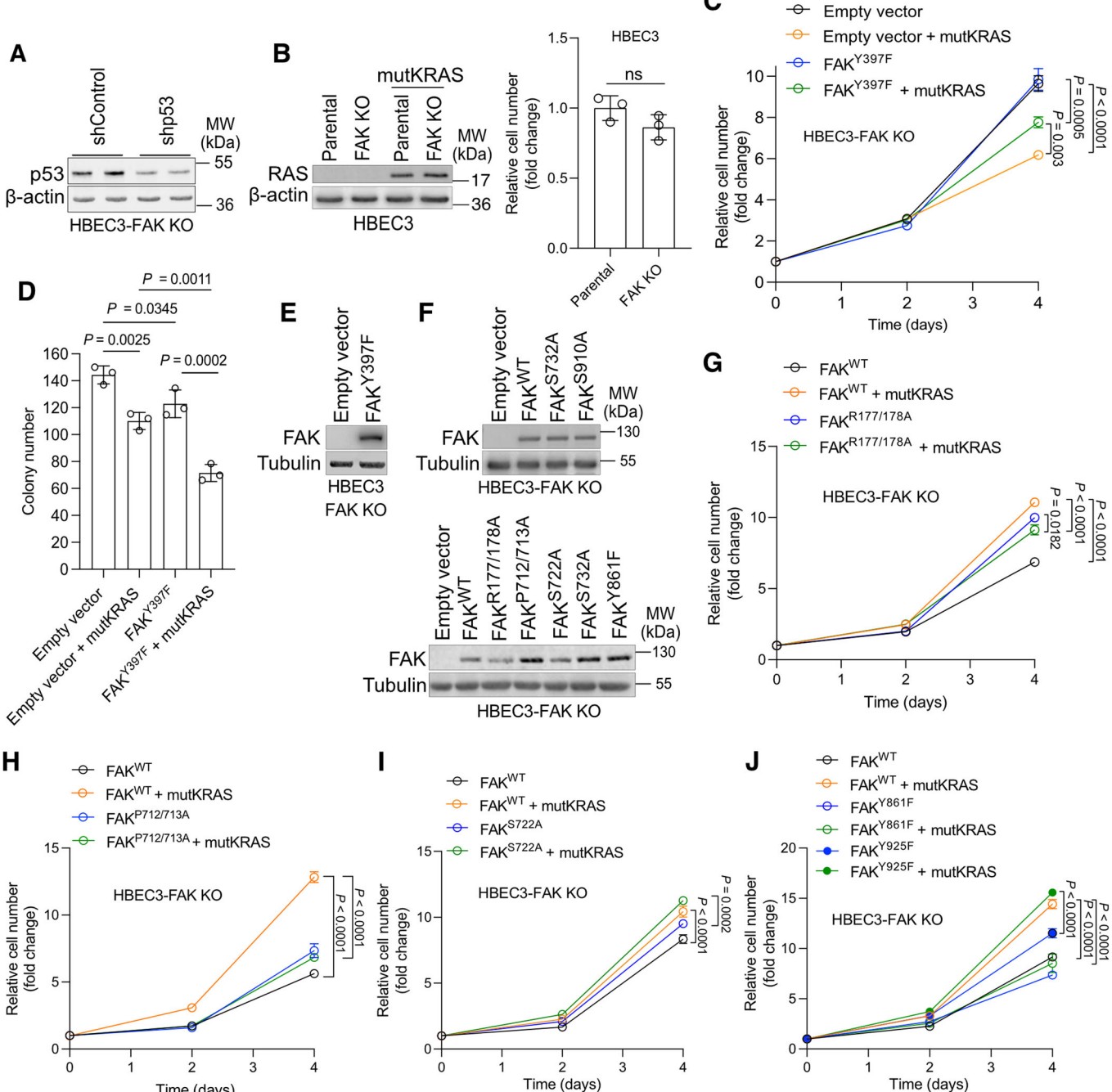

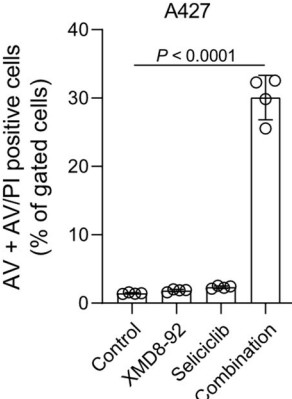
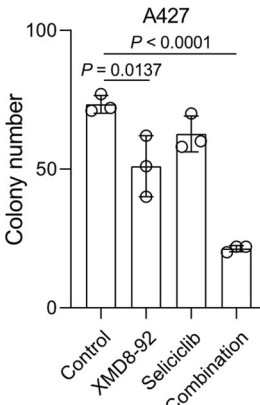

**Figure EV2. Co-inhibition of ERK5 and CDK5 increases apoptosis and suppresses colony forming capacity of KRAS mutant NSCLC cells.**

Relative quantification of cell death by flow cytometry analysis of Annexin V-Atto 633 (AV) + Annexin V/PI (AV/PI)-positive (left) and colony number (right) of A427 cells treated with XMD8-92 or Seliciclib (10 μM for apoptosis assay and 2.5 μM for colony formation each, respectively) alone or in combination; $n = 3$. Graphical data are mean ± SD. Statistical analyses were done using one-way ANOVA; $n$, number of biologically independent samples.

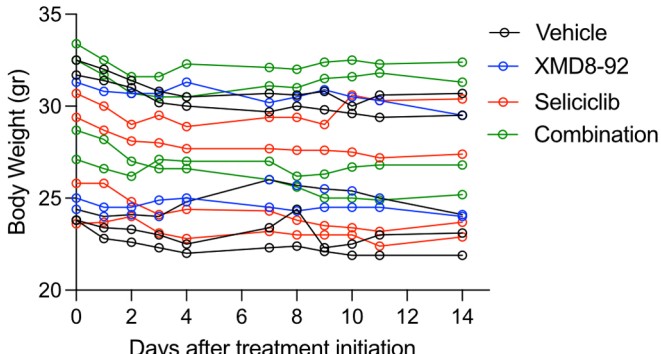

**Figure EV3. The treatment of *LSL-Kras^{G12D/WT};p53^{flox/flox}* mice with XMD8-92 and/or Seliciclib is well tolerated.**

Body weight of *Kras^{G12D/WT};p53^{flox/flox}* mice treated with vehicle or XMD8-92 or Seliciclib or combination of XMD8-92 and Seliciclib for 2 weeks.

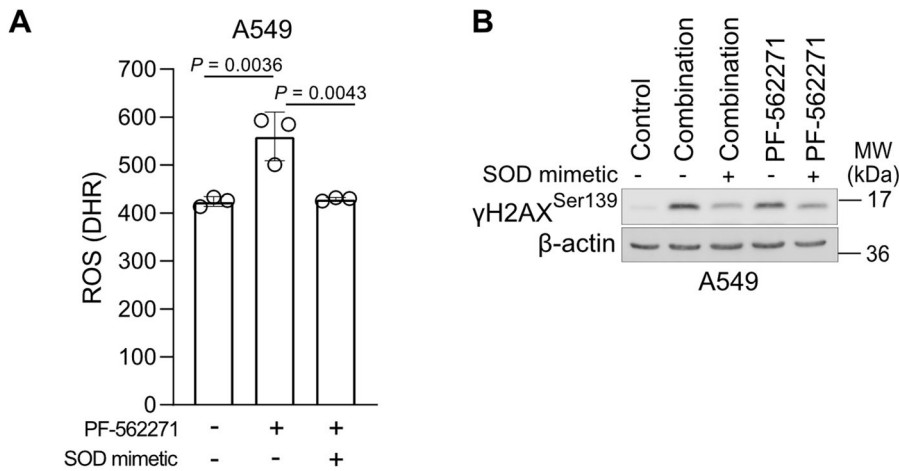

**A** A549

ROS (DHR)

P = 0.0036
P = 0.0043

PF-562271    −    +    +
SOD mimetic    −    −    +

**B**

Control
Combination
Combination
PF-562271
PF-562271

SOD mimetic    −    −    +    −    +    MW (kDa)

γH2AX^Ser139 — 17
β-actin — 36

A549

**C**

-Log Pvalue

0  4.34 ———————— 40

| GO | Description |
|---|---|
| GO:0006260 | DNA replication |
| R-HSA-69278 | Cell Cycle, Mitotic |
| GO:0051052 | regulation of DNA metabolic process |
| GO:0006520 | amino acid metabolic process |
| GO:0140053 | mitochondrial gene expression |
| GO:0006325 | chromatin organization |
| GO:0022613 | ribonucleoprotein complex biogenesis |
| GO:0006259 | DNA metabolic process |
| GO:0071826 | ribonucleoprotein complex subunit organization |
| R-HSA-8953854 | Metabolism of RNA |
| GO:2000278 | regulation of DNA biosynthetic process |
| CORUM:5380 | TRBP containing complex |
| GO:0006403 | RNA localization |
| M66 | PID MYC ACTIV PATHWAY |
| R-HSA-69190 | DNA strand elongation |
| GO:0006913 | nucleocytoplasmic transport |
| WP4290 | Metabolic reprogramming in colon cancer |
| R-HSA-73894 | DNA Repair |
| R-HSA-69239 | Synthesis of DNA |
| GO:1903311 | regulation of mRNA metabolic process |
| GO:0033044 | regulation of chromosome organization |
| GO:0044770 | cell cycle phase transition |
| CORUM:2755 | 17S U2 snRNP |
| R-HSA-379716 | Cytosolic tRNA aminoacylation |
| GO:0006413 | translational initiation |
| R-HSA-6791226 | Major pathway of rRNA processing in the nucleolus and cytosol |
| M14 | PID AURORA B PATHWAY |
| WP4016 | DNA IR-damage and cellular response via ATR |
| R-HSA-9609507 | Protein localization |
| R-HSA-3700989 | Transcriptional Regulation by TP53 |

| GO | Description |
|---|---|
| GO:0040008 | regulation of growth |
| GO:0045785 | positive regulation of cell adhesion |
| GO:0007507 | heart development |
| GO:0031589 | cell-substrate adhesion |
| GO:0001667 | ameboidal-type cell migration |
| GO:0030198 | extracellular matrix organization |
| GO:0048729 | tissue morphogenesis |
| GO:0007167 | enzyme-linked receptor protein signaling pathway |
| WP2572 | Primary focal segmental glomerulosclerosis (FSGS) |
| GO:0035239 | tube morphogenesis |
| hsa04510 | Focal adhesion |
| R-HSA-1474244 | Extracellular matrix organization |
| R-HSA-8957275 | Post-translational protein phosphorylation |
| GO:0098609 | cell-cell adhesion |
| GO:0034330 | cell junction organization |
| GO:0010810 | regulation of cell-substrate adhesion |
| hsa05205 | Proteoglycans in cancer |
| GO:0030036 | actin cytoskeleton organization |
| GO:0009611 | response to wounding |

Cluster 1  Cluster 2  Cluster 3  Cluster 4  Cluster 5  undefined

◀ **Figure EV4. SOD mimetics rescue ROS-induced DNA damage in mutant KRAS NSCLC.**

(A) Quantification of DHR (ROS marker, green) in A549 cells treated with PF-562271 (10 µM) in the presence or absence of the SOD mimetic, MnTMPyp (25 µM); $n = 3$. Graphical data are mean ± SD. Statistical analyses were done using one-way ANOVA; $n$, number of biologically independent samples. (B) Immunoblot analysis for the indicated targets in A549 cells line, treated with DMSO (control) or with a combination of XMD8-92 and Seliciclib (10 µM) or PF-562271 (5 µM) in the presence or absence of the SOD mimetic, MnTMPyp (25 µM). (C) Metascape-derived analysis of the functional categories associated to the 5 clusters from main Fig. 5C. Enriched terms were filtered based on the enrichment score and accumulative hypergeometric $P$ values ($P < 0.05$). Remaining significant terms were then hierarchically clustered into a tree based on Kappa-statistical similarities among their gene memberships. Then 0.3 kappa score was applied as the threshold to cast the tree into term clusters.

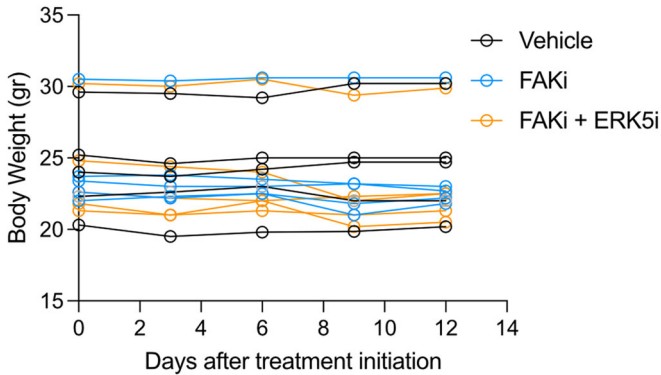

**Figure EV5. The treatment of *LSL-Kras*^{G12D/WT};*p53*^{flox/flox} mice with VS4718 (FAKi) or VS4718 (FAKi) + XMD8-92 (ERK5i) is well tolerated.**

Body weight of *Kras*^{G12D/WT};*p53*^{flox/flox} mice treated with vehicle or VS-4718 (FAKi) or a combination of VS-4718 and XMD8-92 (FAKi + ERK5i) for 2 weeks.

