## [Peer Review File · EMBO Molecular Medicine]

ERK5 suppression overcomes FAK inhibitor resistance in mutant KRAS-driven non-small cell lung cancer

Georgia Konstantinidou, Chiara Pozzato, Gonçalo Outeiro-Pinho, Mirco Galiè, and Giorgio Ramadori

Corresponding author: Georgia Konstantinidou (georgia.konstantinidou@unibe.ch)

Review Timeline:

Submission Date: Editorial	2nd Nov 23
Decision:	6th Nov 23
Authors' Correspondence:	6th Nov 23
Editor's Correspondence:	6th Nov 23
Appeal:	13th Feb 24
Editorial Decision:	19th Mar 24
Revision Received:	26th Jul 24
Editorial Decision:	20th Aug 24
Revision Received:	28th Aug 24
Accepted:	29th Aug 24

Editor: *Zeljko Durdevic*

Transaction Report:

6th Nov 2023

Decision on your manuscript EMM-2023-18934

Dear Prof. Konstantinidou,

Thank you for the submission of your research manuscript to our editorial offices. I have now had the opportunity to read it and to discuss it with the other members of our editorial team. I am afraid we all agree that the manuscript is not well suited for publication in EMBO Molecular Medicine and have therefore decided not to proceed with its handling and peer review.

The study identifies ERK5 as a mediator of FAK resistance in mutant KRAS-driven non-small cell lung cancer. We do recognize potential interest the findings, however, translational implications of your work are not further developed, limiting the overall clinical insights that are key for publication in EMBO Molecular Medicine. In particular, ERK5 inhibition as a potential co-targeting strategy to counteract FAK inhibitor resistance in NSCLC requires further investigation in an appropriate in vivo model. Therefore, I am afraid that we cannot offer further consideration to your article.

That being said, I discussed your work with editors at our sister journal Life Science Alliance (www.life-science-alliance.org). Life Science Alliance is an open access journal launched in partnership between EMBO, Rockefeller, and Cold Spring Harbor Laboratory Presses, and it publishes work that is of high value to the respective communities across all areas in the life sciences. I am glad to say that Life Science Alliance is interested in considering your work for publication and would like to send it out for formal peer-review.

I am very sorry to disappoint you on this occasion and I hope you will view the encouragement of a transfer favorably. If this is the case, please use the link below to transfer the manuscript directly; no re-formatting is required.

Yours sincerely,

Zeljko Durdevic

=====

As a service to authors, EMBO provides authors with the possibility to transfer a manuscript that one journal cannot offer to publish to another EMBO publication. The full manuscript and if applicable, reviewers reports are automatically sent to the receiving journal to allow for fast handling and a prompt decision on your manuscript. For more details of this service, and to transfer your manuscript to another EMBO title please click on Link Not Available

Dear Dr Durdevic,

Thank you for evaluating our manuscript. We are currently performing the co-targeting of ERK5 and FAK in FAK-resistant lung tumors in our lung adenocarcinoma mouse model. Would that be sufficient to accept a resubmission of our manuscript and have it send out for review?

Thanks a lot in advance,

Best regards,

Georgia

Dear Georgia,

Thank you for the inquiry. We would welcome resubmission of the manuscript with the co-targeting of ERK5 and FAK in FAK-resistant lung tumors in a lung adenocarcinoma mouse model. Of course, if the results support potential clinical translation.

Best wishes,

Zeljko

19th Mar 2024

Dear Prof. Konstantinidou,

Thank you for the submission of your manuscript to EMBO Molecular Medicine, and please accept my apologies for the delay in getting back to you, which is due to the fact that one referee needed more time to complete his/her review. We have now received feedback from the three reviewers who agreed to evaluate your manuscript.

As you will see from the reports pasted below, while the referee #2 supports publication of the manuscript, referees #1 and #3 recognize potential interest and clinical relevance of the study but also raise important concerns that should be addressed in a major revision of the current manuscript. Addition of experiments using a human LUAD xenograft model as suggested by the referee #3 is encouraged, however and as mentioned by the referee (point 5) this can also be discussed as a limitation of the study. If you would like to discuss further the points raised by the referees, I am available to do so via email or video. Let me know if you are interested in this option.

We would welcome the submission of a revised version within three to six months for further consideration. Please let us know if you require longer to complete the revision.

I look forward to receiving your revised manuscript.

Yours sincerely,

Zeljko Durdevic

We require:

- 1) A .docx formatted version of the manuscript text (including legends for main figures, EV figures and tables). Please make sure that the changes are highlighted to be clearly visible.
- 2) Individual production quality figure files as .eps, .tif, .jpg (one file per figure). For guidance, download the 'Figure Guide PDF': (<https://www.embopress.org/page/journal/17574684/authorguide#figureformat>).
- 3) A .docx formatted letter INCLUDING the reviewers' reports and your detailed point-by-point responses to their comments. As part of the EMBO Press transparent editorial process, the point-by-point response is part of the Review Process File (RPF), which will be published alongside your paper.
- 4) A complete author checklist, which you can download from our author guidelines (<https://www.embopress.org/page/journal/17574684/authorguide#submissionofrevisions>). Please insert information in the checklist that is also reflected in the manuscript. The completed author checklist will also be part of the RPF.

6) It is mandatory to include a 'Data Availability' section after the Materials and Methods. Before submitting your revision, primary datasets produced in this study need to be deposited in an appropriate public database, and the accession numbers and database listed under 'Data Availability'. Please remember to provide a reviewer password if the datasets are not yet public (see <https://www.embopress.org/page/journal/17574684/authorguide#dataavailability>).

13) Author contributions: You will be asked to provide CRediT (Contributor Role Taxonomy) terms in the submission system. These replace a narrative author contribution section in the manuscript.

14) A Conflict of Interest statement should be provided in the main text.

15) Every published paper now includes a 'Synopsis' to further enhance discoverability. Synopses are displayed on the journal webpage and are freely accessible to all readers. They include a short stand first (maximum of 300 characters, including space) as well as 2-5 one-sentence bullet points that summarize the paper. Please write the bullet points to summarize the key NEW findings. They should be designed to be complementary to the abstract - i.e. not repeat the same text. We encourage inclusion of key acronyms and quantitative information (maximum of 30 words / bullet point). Please use the passive voice. Please attach these in a separate file or send them by email, we will incorporate them accordingly.

Please also suggest a striking image or visual abstract to illustrate your article as a PNG file 550 px wide x 300-800 px high.

**** Reviewer's comments ****

Referee #1 (Comments on Novelty/Model System for Author):

Results are novel, and experiments performed with high technical quality and adequate model systems. Results could have clinical impact to treat patients with lung cancer.

Referee #1 (Remarks for Author):

In this manuscript, Pozzato and colleagues investigate possible mechanisms that lead to FAK inhibitor resistance in KRAS mutant LUAD. In particular, the authors identify ERK5 as mediator of FAK inhibitor resistance providing a possible new therapeutic strategy for patients with LUAD. The research is original, of potential therapeutic value and the experiments are adequately performed. The results are convincing and nicely add to our comprehension of important and potentially targetable mechanisms in mutant Kras tumors. I have a few specific comments:

1. Figure 1: there is no validation of p53 knockdown, which should be assessed by WB. Because p53 expression is generally low in cells, the efficiency of knockdown should be tested in p53-stimulating conditions. Also, reviewer lacks information about the strategy used to knock down p53. If this was through shRNA, how many different shRNAs were used?
2. In a previous study by the authors (Konstantinidou, 2013), Y397 was found as an important phosphorylated residue for FAK function in Kras mutant lung tumor cells. How would a FAK Y397F mutant compare to the other mutations studied here, in terms of tumor cell proliferation / colony formation as shown in Fig. 1G-J?
3. The effect of blocking together ERK5 and CDK5 pharmacologically on tumor development agrees with the importance of the identified residues, S732 and S910 on FAK. Thus, what would be the result for the expression of combined mutations in these residues (S732A,S910A) for HBEC-FAK KO cell proliferation and colony formation, as reported in Fig. 1G-J?
4. To establish the connection between ERK5, CDK5 and FAK more firmly, a FAK KO (using CRISPR/Cas9 like the one used in HBEC cells for Fig. 1) should be done in the used tumor cells, A549, A427 and H460. In these FAK KO conditions, what would be the effect of combined XMD8-92 and seliciclib compared to what was obtained in FAK expressing cells (Fig. 2A)? In addition, or an alternative would be to generate stable cells expressing either FAK S732A,S910A or FAK S732E,S910E to mimic non-phosphorylatable or, respectively, constitutively phosphorylated FAK, and to test the action of the drugs in this context.

Minor comments:

- Specify the time point used in Figure S1A for relative cell number count. Was it calculated at day 2 or 4? To be in line with the other figures of the paper, it would be better to provide proliferation curves with both times, or at least at 4 days.
- Did the mice show any side effects due to the treatments?
- Reviewer feels that treating KP mice at 12 weeks post-tumor initiation is not treating late-stage tumors, thus this should be removed from the text (p. 11). Unless staging is actually performed.
- In the Discussion, the authors should add information about the therapeutic window such combination treatments could have or not in the clinics. Are toxicities expected?

- Because kinase inhibitors are never specific, is there information about the selectivity of these three compounds (ERKi, FAKi, CDK5i) and potential off-targets known from the literature? This should be discussed.

Referee #2 (Comments on Novelty/Model System for Author):

Overall, this manuscript identifies two new Focal Adhesion Kinase (FAK) interactors, namely ERK5 and CDK5, as relevant players in FAK-dependent maintenance of KRAS mutant NSCLC. The biochemical data and the in vitro and in vivo experiments are well conducted. The results are reliable and innovative.

Referee #2 (Remarks for Author):

In this manuscript the authors identify two new Focal Adhesion Kinase (FAK) interactors, namely ERK5 and CDK5, as relevant players in FAK-dependent maintenance of KRAS mutant NSCLC. The results are well described, starting from the identification of FAK serine phosphorylation 732 and 910 as critical for the proliferation of mutant KRAS-transformed lung cells. These two sites can be phosphorylated by CDK5 and ERK5, respectively. They show that inhibition of ERK5 and CDK5 synergistically suppress FAK function, proliferation and survival of NSCLC cells. Moreover they also demonstrate that the pharmacological inhibition of ERK5 and CDK5 inhibits tumor progression in a mouse model of KrasG12D-driven lung adenocarcinoma. Finally, pharmacological inhibition of ERK5 combined to FAK inhibition has the potential to overcome resistance to FAK inhibitors, enhancing the antitumor response.

Overall, the data presented are solid, based on biochemical and cellular assays, going from the in vitro to the in vivo conditions. The data are highly significant.

Referee #3 (Remarks for Author):

EMM-2023-18934-V2-Q

Pozzato et al. "ERK5 suppression overcomes FAK inhibitor resistance in mutant KRAS driven non-small cell lung cancer "

The authors study the role of development of resistance to FAK inhibition in KRAS mutant preclinical models of human lung cancer using genetically manipulated immortalized normal human lung epithelial cells (HBECs), patient derived tumor cell lines to identify two "interactors" ERK5 and CDK5 as potential therapeutic targets. Using the HBEC model system they show that in the presence of oncogenic KRAS ERK5 and CDK5 are required to provide resistance to FAK genetic depletion or pharmacologic inhibition. Importantly inhibition of ERK5 and CDK5 suppress FAK function leading to programmed cell death resulting from "ROS-induced DNA damage." In a genetically engineered mouse model (GEMM) of KRAS lung adenocarcinoma (LUAD) drug targeting of ERK5 and CDK5 inhibited tumor formation in vivo. ERK5 pharmacologic inhibition prevented development of FAK inhibitor resistance with increased anti-tumor responses in preclinical studies. They conclude: " Therefore, we propose ERK5 inhibition as a potential co-targeting strategy to counteract FAK inhibitor resistance in NSCLC. "

Comments to the authors:

The manuscript is reviewed in the context that FAK inhibition has shown dramatic results in KRAS driven preclinical models of LUAD, and while it has been tested in the clinic with some indication of effect, resistance has universally developed. If ways, especially pharmacologic approaches, to enhance FAK inhibitory treatment could be developed this would generate immediate clinical trials given the limitations of even our best oncogenic KRAS targeting drugs. It is also reviewed in the context of the knowledge of FAK inhibition in LUAD leading to alterations in DNA repair (e.g. after radiation, see authors Tang et al. 2016 citation), the massive efforts going into develop CDK5 inhibitors for other diseases (such as neurologic function), and what is known about the specificity of ERK5 and CDK5 available inhibitors.

1. The work is all technically well done and clearly presented. The interactions of ERK5 and CDK5 with FAK and the potential for combined targeting of ERK5/CDK5 (alone or together) to overcome FAK inhibitor resistance in KRAS mutant LUAD is clear.
2. The first major issue is that there is no human tumor xenograft data presented. While the GEMM in vivo data are interesting an important, clearly the approach did not appear to be curative (remove all tumors). However, equally important, for planning for clinical translation, investigators will want to see anti-tumor activity in human KRAS mutant LUAD xenografts with FAK inhibition

combined with targeting ERK5, CDK5 alone and together. As part of this, given the role of LKB1/STK11 mutations co-occurring with KRAS mutations and the potential for targeting FAK in preclinical models (see JCI Insight 2017 Gilbert-Ross et al. PMC5333956) it would be important to include at least one KRAS mutant LKB1 mutant xenograft in these tests.

3. Similarly, given current clinical translation of targeted therapy in patients for KRASG12C and KRASG12D mutant LUAD, it would be important to know the results comparing targeting oncogenic KRAS, FAK, and ERK5 or CDK5 in one of the human tumor xenograft models.

4. A major issue is the specificity of the two drugs used in these studies - XMD8-92 and inhibitor of ERK5 which also targets BRD4, and seliciclib (roscovitine) which also inhibits CDK1, CDK2, CDK7 in addition to CDK5. The issue here is whether the anti-tumor effects seen with pharmacologic inhibition are due to ERK5 and CDK5 targeting or to one of the other targets these drugs hit. There could be several ways to address this key issue - including showing that BRD4 knockdown does not give the same results as XMD8-92, failure of exogenous BRD4 to "rescue" the XMD8092 effects, the use of some more specific CDK5 inhibitors (e.g. GFB-128-11, or CDK5 inhibitor small peptide). Alternatively, inducible knockdown of ERK5, CDK5 could be generated to be used in vivo in a xenograft that was then treated with the FAK inhibitor to show the effect occurred with functional genomic ablation even if we do not yet have very specific ERK5 or CDK5 inhibitors.

5. All of these issues can also be mentioned in the Discussion section as "limitations of the current study"

6. Given the importance for their studies of identifying additional ERK5 inhibitors with specificity the authors may want to cite some of the current literature (e.g. Modulation of ERK5 Activity as a Therapeutic Anti-Cancer Strategy. Duncan C. Miller, et al., J. Med. Chem. 2023, 66, 4491–4502) including the problem of paradoxical activation of ERK5 by ERK5 inhibitors (e.g. Lochhead et al. Nature Comm 2020.), and the use of ERK5 inhibition to sensitize tumors to TRAIL targeted therapy (Espinosa-Gil et al. CDD, 2023).

Referee #1 (Comments on Novelty/Model System for Author):

Results are novel, and experiments performed with high technical quality and adequate model systems. Results could have clinical impact to treat patients with lung cancer.

Thank you for recognizing the novelty of the findings and technical quality of our experiments.

Referee #1 (Remarks for Author):

In this manuscript, Pozzato and colleagues investigate possible mechanisms that lead to FAK inhibitor resistance in KRAS mutant LUAD. In particular, the authors identify ERK5 as mediator of FAK inhibitor resistance providing a possible new therapeutic strategy for patients with LUAD. The research is original, of potential therapeutic value and the experiments are adequately performed. The results are convincing and nicely add to our comprehension of important and potentially targetable mechanisms in mutant Kras tumors. I have a few specific comments:

1. Figure 1: there is no validation of p53 knockdown, which should be assessed by WB. Because p53 expression is generally low in cells, the efficiency of knockdown should be tested in p53-stimulating conditions. Also, reviewer lacks information about the strategy used to knock down p53. If this was through shRNA, how many different shRNAs were used?

We thank the reviewer for the comments. We used shRNA strategy to knockdown p53. The p53 steady-state level in HBEC cells is enough to be detected by wb without stimulation. We used one shRNA that has been previously validated and published by the lab of Didier Trono (PMID: 16432520). We have added this information in the materials and methods as well as reagents excel file. We have added the wb with p53 knockdown in Supplementary Fig. 1A and below (Fig. 1A) for your perusal.

Figure 1. knockdown of p53 in HBEC3-FAK KO cells. (A) Immunoblot for the indicated targets of HBEC3-FAK KO cells previously transduced with a lentivirus expressing shControl or a shRNA against human p53.

2. In a previous study by the authors (Konstantinidou, 2013), Y397 was found as an important phosphorylated residue for FAK function in Kras mutant lung tumor cells. How would a FAK Y397F mutant compare to the other mutations studied here, in terms of tumor cell proliferation / colony formation as shown in Fig. 1G-J?

In fact, the reviewer is right and the mutant FAK^{Y397F} was our “positive control” for the mutations experiments. The effect of FAK^{Y397F} mutant closely recapitulates the effect of the empty vector (FAK KO) in the presence of mutKRAS as they both reduced cell proliferation and colony forming capacity in the presence of mutKRAS, please see Fig. 2A-2C below (manuscript Fig. S1C and S1E).

Figure 2. Expression of a loss-of-function point mutant of FAK at position 397 (FAK^{Y397F}), reduces cell proliferation and colony forming capacity in presence of mutKRAS. (A) Immunoblot for the indicated targets of HBEC FAK KO cells transduced with a retrovirus expressing empty vector (pWZL hygro) or a vector carrying FAK^{Y397F} followed by transduction with a plasmid expressing mutKRAS. (B) Relative cell number of HBEC FAK KO cells transduced as described in (A). (C) Colony formation of HBEC FAK KO cells transduced as described in (A).

3. The effect of blocking together ERK5 and CDK5 pharmacologically on tumor development agrees with the importance of the identified residues, S732 and S910 on FAK. Thus, what would be the result for the expression of combined mutations in these residues (S732A,S910A) for HBEC-FAK KO cell proliferation and colony formation, as reported in Fig. 1G-J?

We thank the reviewer for suggesting to perform this experiment. We mutagenized the human WT FAK to generate the S732A,S910A double mutation. However, we noticed by immunoblot that the protein levels of FAK upon ectopic expression of S732A/S910A double mutant plasmid was lower compared to the WT FAK plasmid expression and other FAK single point mutants FAK (Y397F) (Fig. 3A and 3B below), which was not the case for other mutants as shown in the manuscript Fig. S1F of the manuscript. We think that this double mutation may somehow affect the overall stability of FAK, which needs careful further investigation and goes beyond the scope of this manuscript. Of note, this was not the case for other FAK point mutations or drug treatments as the total FAK protein levels was unchanged (Figs. S1F and main fig. 2F). Accordingly, ectopic expression of mutKRAS in the S732A/S910A double mutant FAK expressing cells led to reduced proliferation (Fig. 3C below). However, the reduced proliferation might be due to the low total FAK levels and not due to the lower pTyr397-FAK protein levels. We report these results below for the reviewer’s perusal only.

Figure 3. Expression of a double loss-of-function point mutant of FAK at positions 732 and 910 (FAK^{S732A/S910A}) reduces the stability of FAK. (A) Immunoblot for the indicated targets of HBEC FAK KO cells transduced with a retrovirus expressing wild type (WT) FAK or a vector carrying FAK^{S732A/S910A}. **(B)** Immunoblot for the indicated targets of HBEC FAK KO cells transduced with a retrovirus expressing an empty vector (pWZL hygro) or a vector carrying FAK^{S732A/S910A} or a vector carrying a loss-of-function point mutant of FAK, FAK^{Y397F}. **(C)** Relative cell number of HBEC FAK KO cells transduced with FAK^{S732A/S910A} and with a plasmid expressing mutKRAS.

4. To establish the connection between ERK5, CDK5 and FAK more firmly, a FAK KO (using CRISPR/Cas9 like the one used in HBEC cells for Fig. 1) should be done in the used tumor cells, A549, A427 and H460. In these FAK KO conditions, what would be the effect of combined XMD8-92 and seliciclib compared to what was obtained in FAK expressing cells (Fig. 2A)? In addition, or an alternative would be to generate stable cells expressing either FAK S732A,S910A or FAK S732E,S910E to mimic non-phosphorylatable or, respectively, constitutively phosphorylated FAK, and to test the action of the drugs in this context.

We thank the reviewer for the comments. As FAK is a synthetic vulnerability of KRAS mutant cancer cells, we are unable to knockout FAK via CRISPR/Cas9 in NSCLC cells already harbouring KRAS mutations (A549, A427 and H460) to generate stable cell lines as this procedure requires time and they don't survive enough for follow-up studies. This is why we engineered the HBEC cell line and built up step-by-step the genetic alterations to study the dependency of these cells on FAK (in these experiments we first knocked out FAK and then we expressed mutKRAS). We attempted to use the cell line that we generated as part of the comment/answer #3 above (HBEC-FAK KO cells harbouring S732A/S910A), but they are also not healthy after mutKRAS expression which includes 3 weeks of selection, precluding additional cytotoxic experiments with drugs on top of it.

Minor comments:

- Specify the time point used in Figure S1A for relative cell number count. Was it calculated at day 2 or 4? To be in line with the other figures of the paper, it would be better to provide proliferation curves with both times, or at least at 4 days.

Thank you for the comment. The proliferation on the panel in S1A (now appears as S1B) was calculated at day 4 as the other figures. We have added this information in the figure legend.

- Did the mice show any side effects due to the treatments?

The mice in the XMD8-92/ Seliciclib cohort all lost some weight at the beginning (up to almost 8%) until day 4 after starting the treatment, including the vehicle control suggesting a vehicle effect rather than toxicity due to the drugs. See panel A in Fig. below and manuscript Fig. S3A.

The XMD8-92/VS4718 cohort did not show any reduction in body weight when given alone or any other parameters based on our score sheets. However, during the combination of XMD8-92 + VS4718 treatment 3 mice showed a moderate reduction in body weight (up to almost 8%.) at day 6-9. After that, some regained weight or remained stable until the end of the treatment. See panel B below and Fig. S6A of the new manuscript.

Figure 4. Body weights of mice during treatments with the indicated pharmacological inhibitors. (A) Treatment with XMD8-92, seliciclib or combination of both. **(B)** Treatment with VS4718, XMD8-92 or combination of both drugs.

- Reviewer feels that treating KP mice at 12 weeks post-tumor initiation is not treating late-stage tumors, thus this should be removed from the text (p. 11). Unless staging is actually performed.

We agree and have eliminated this sentence from the manuscript.

- In the Discussion, the authors should add information about the therapeutic window such combination treatments could have or not in the clinics. Are toxicities expected?

Based on our preclinical trial data, the combination was well tolerated despite the i.p. injections and vehicle that seemed to moderately affect the XMD8-92/Seliciclib cohorts (refer to the above point of the body weights). As Seliciclib is orally available we expect that they are even better tolerated in patients. Indeed, Seliciclib is very well tolerated as single drug. Moreover, it has also been tested in a Phase I trial in combination with Gemcitabine and Cisplatin in NSCLC patients. The maximum tolerated dose was 800 mg in combination with 1000 mg/m² Gemcitabine and 75 mg/m² Cisplatin. Interestingly, for this drug combination the level of haematological toxicity observed was low (PMID: 20822897). Unfortunately, there are no MEK5/ERK5 inhibitors in clinical testing at the moment. Therefore, it is difficult to predict the therapeutic window that these combination therapies may have in the clinic. We have discussed these issues in the discussion section.

- Because kinase inhibitors are never specific, is there information about the selectivity of these three compounds (ERKi, FAKi, CDK5i) and potential off-targets known from the literature? This should be discussed.

We agree with the statement that kinase inhibitors are never specific and this is also the case with the inhibitors used in this study namely, XMD8-92 (inhibits also BRD4 and LRRK2), Seliciclib (inhibits also other CDKs) and VS4718 (PYK2). It is however, important to note that XMD8-92 and Seliciclib based on our results in lung cancer don't impact cell proliferation or cell death when used alone. This is also confirmed with the use of shRNAs (manuscript Fig. 2G-2I). This suggests that any single drug off-target effect is not functionally important in this context. We only observed anti-proliferative and pro-apoptotic effects when we combined these drugs and shRNAs together. Moreover, from our experiments in Fig. 4, we could pinpoint that the functional consequences of the combination of XMD8-92 and Seliciclib, which is identical with that of the FAK inhibitor, are restricted to the induction of ROS and DNA damage, as the cell proliferation defect was totally rescued by the antioxidant compound SOD2. Altogether, these results suggest that the known off-target effects of these compounds may not be relevant in the context of lung cancer. We have discussed these issues in the discussion section as a limitation of our studies.

Referee #2 (Comments on Novelty/Model System for Author):

Overall, this manuscript identifies two new Focal Adhesion Kinase (FAK) interactors, namely ERK5 and CDK5, as relevant players in FAK-dependent maintenance of KRAS mutant NSCLC. The biochemical data and the in vitro and in vivo experiments are well conducted. The results are reliable and innovative.

Referee #2 (Remarks for Author):

In this manuscript the authors identify two new Focal Adhesion Kinase (FAK) interactors, namely ERK5 and CDK5, as relevant players in FAK-dependent maintenance of KRAS mutant NSCLC. The results are well described, starting from the identification of FAK serine phosphorylation 732 and 910 as critical for the proliferation of mutant KRAS-transformed lung cells. These two sites can be phosphorylated by CDK5 and ERK5, respectively. They show that inhibition of ERK5 and CDK5 synergistically suppress FAK function, proliferation and survival of NSCLC cells. Moreover they also demonstrate that the pharmacological inhibition of ERK5 and CDK5 inhibits tumor progression in a mouse model of KrasG12D-driven lung adenocarcinoma. Finally, pharmacological inhibition of ERK5 combined to FAK inhibition has the potential to overcome resistance to FAK inhibitors, enhancing the antitumor response.

Overall, the data presented are solid, based on biochemical and cellular assays, going from the in vitro to the in vivo conditions. The data are highly significant.

Thank you for recognizing the significance of our findings and technical quality of our experiments.

Referee #3 (Remarks for Author):

EMM-2023-18934-V2-Q

Pozzato et al. "ERK5 suppression overcomes FAK inhibitor resistance in mutant KRAS driven non-small cell lung cancer "

The authors study the role of development of resistance to FAK inhibition in KRAS mutant preclinical models of human lung cancer using genetically manipulated immortalized normal human lung epithelial cells (HBECs), patient-derived tumor cell lines to identify two "interactors" ERK5 and CDK5 as potential therapeutic targets. Using the HBEC model system they show that in the presence of oncogenic KRAS ERK5 and CDK5 are required to provide resistance to FAK genetic depletion or pharmacologic inhibition. Importantly inhibition of ERK5 and CDK5 suppress FAK function leading to programmed cell death resulting from "ROS-induced DNA damage." In a genetically engineered mouse model (GEMM) of KRAS lung adenocarcinoma (LUAD) drug targeting of ERK5 and CDK5 inhibited tumor formation in vivo. ERK5 pharmacologic inhibition prevented development of FAK inhibitor resistance with increased anti-tumor responses in preclinical studies. They conclude: " Therefore, we propose ERK5 inhibition as a potential co-targeting strategy to counteract FAK inhibitor resistance in NSCLC. "

Comments to the authors:

The manuscript is reviewed in the context that FAK inhibition has shown dramatic results in KRAS driven preclinical models of LUAD, and while it has been tested in the clinic with some indication of effect, resistance has universally developed. If ways, especially pharmacologic approaches, to enhance FAK inhibitory treatment could be developed this would generate immediate clinical trials given the limitations of even our best oncogenic KRAS targeting drugs. It is also reviewed in the context of the knowledge of FAK inhibition in LUAD leading to alterations in DNA repair (e.g. after radiation, see authors Tang et al. 2016 citation), the massive efforts going into develop CDK5 inhibitors for other diseases (such as neurologic function), and what is known about the specificity of ERK5 and CDK5 available inhibitors.

1. The work is all technically well done and clearly presented. The interactions of ERK5 and CDK5 with FAK and the potential for combined targeting of ERK5/CDK5 (alone or together) to overcome FAK inhibitor resistance in KRAS mutant LUAD is clear.

We thank the reviewer for recognizing the quality of our work.

2. The first major issue is that there is no human tumor xenograft data presented. While the GEMM in vivo data are interesting an important, clearly the approach did not appear to be curative (remove all tumors). However, equally important, for planning for clinical translation, investigators will want to see anti-tumor activity in human KRAS mutant LUAD xenografts with FAK inhibition combined with targeting ERK5, CDK5 alone and together.

We thank the reviewer for the suggestions. As the reviewer stated in point #5 we agree that these above points could be added as study limitations due to the use of an already high

number of animals used for our study. Given the general relevance of the KRAS-driven GEMM model in recapitulating the tumor microenvironment of lung cancer and predict response to therapy we are confident that the treatment combination will have an impact in the clinic.

The reviewer stated that “our approach did not appear to be curative”. I would like to add here that we evaluated the impact of this therapy at the end of the 2 weeks treatment. Even with this treatment scheme we observed a high percentage of tumor cell apoptosis induction in all mice treated with FAKi + ERK5i out of which one mouse had no detectable tumors (Fig. 6D and 6F). Therefore, the results show a trend towards tumor regression.

As part of this, given the role of LKB1/STK11 mutations co-occurring with KRAS mutations and the potential for targeting FAK in preclinical models (see JCI Insight 2017 Gilbert-Ross et al. PMC5333956) it would be important to include at least one KRAS mutant LKB1 mutant xenograft in these tests.

The human cell lines A549, A427 and H460 that we used in vitro harbour a deletion in *STK11*. Therefore, our findings (development of resistance upon FAKi and response to ERK5i and CDK5i), at least based on our in vitro observations, are expected to also apply for the *STK11* deleted tumors that co-occur with KRAS mutations.

3. Similarly, given current clinical translation of targeted therapy in patients for KRASG12C and KRASG12D mutant LUAD, it would be important to know the results comparing targeting oncogenic KRAS, FAK, and ERK5 or CDK5 in one of the human tumor xenograft models.

We thank the reviewer for this suggestion. We agree that comparing the efficacy of these 3 treatments is crucial. Despite the FDA approval of highly specific inhibitors targeting KRAS^{G12C} (sotorasib and adagrasib) in the clinic, clinical trial data evidence that not all patients show partial response, owing to the development of drug resistance (Awad, Liu et al. 2021). Notably, recent findings indicated that non-genetic acquired resistance to sotorasib involves alterations in focal adhesions (Mohanty, Nam et al. 2023). In fact, we plan to perform these experiments, including to assess whether in KRAS^{G12C} inhibitor-resistant cells and tumors in mice, FAK and ERK5 targeting may be also effective therapeutic strategy both to potentiate the efficacy of KRAS inhibitors and to prevent the development of non-genetic acquired drug resistance. However, the generation of necessary tools and the correct execution of these experiments requires time that goes beyond the scope of the current manuscript. As suggested, we have discussed this in the discussion section.

4. A major issue is the specificity of the two drugs used in these studies - XMD8-92 and inhibitor of ERK5 which also targets BRD4, and seliciclib (roscovitine) which also inhibits CDK1, CDK2, CDK7 in addition to CDK5. The issue here is whether the anti-tumor effects seen with pharmacologic inhibition are due to ERK5 and CDK5 targeting or to one of the other targets these drugs hit. There could be several ways to address this key issue - including showing that BRD4 knockdown does not give the same results as XMD8-92, failure of exogenous BRD4 to "rescue" the XMD8092 effects, the use of some more specific CDK5 inhibitors (e.g. GFB-128-11, or CDK5 inhibitor small peptide). Alternatively, inducible knockdown of ERK5, CDK5 could be generated to be used in vivo in a xenograft that was then treated with the FAK inhibitor to show the

effect occurred with functional genomic ablation even if we do not yet have very specific ERK5 or CDK5 inhibitors.

In our studies in vitro, when we used XMD8-92 or Seliciclib alone, we did not observe a biologically significant antitumoral effect (Fig. 2A-2D). Contrary, we observed a potent anti-proliferative and pro-apoptotic effect only when we combined these 2 inhibitors. This suggests that direct inhibition of the off-targets (BRD4 or CDK1/2/7) does not have any impact on the proliferation or survival of lung cancer cells. Moreover, in Fig. 2G-2H we showed that shRNAs against ERK5 and CDK5 can recapitulate the anti-proliferative and pro-apoptotic effects of XMD8-92 and seliciclib in vitro. We agree that the limitation of our study is the lack of xenografts with inducible knockdown of ERK5, CDK5. We have discussed this in the text of Fig. 2 and in the discussion section.

5. All of these issues can also be mentioned in the Discussion section as "limitations of the current study".

As mentioned above, we have discussed these issues in the discussion session.

6. Given the importance for their studies of identifying additional ERK5 inhibitors with specificity the authors may want to cite some of the current literature (e.g. Modulation of ERK5 Activity as a Therapeutic Anti-Cancer Strategy. Duncan C. Miller, et al., J. Med. Chem. 2023, 66, 4491–4502) including the problem of paradoxical activation of ERK5 by ERK5 inhibitors (e.g. Lochhead et al. Nature Comm 2020.), and the use of ERK5 inhibition to sensitize tumors to TRAIL targeted therapy (Espinosa-Gil et al. CDD, 2023).

As suggested, we have included these references in the results and discussion sections.

20th Aug 2024

Dear Prof. Konstantinidou,

Thank you for the submission of your revised manuscript to EMBO Molecular Medicine. I am pleased to inform you that we will be able to accept your manuscript pending the following final amendments:

1) Figures:

- Please reduce number of EV figures to max. 5, e.g. by fusing 2 or more EV figures with one panel. Also, in figures with one panel labeling "A" should be omitted. Please update the callouts in the text accordingly.
- Please correct EV figure nomenclature to "Figure EV1" etc. in the legends.

2) In the main manuscript file, please do the following:

- Please address all comments suggested by our data editors listed below:

o Data availability statement:

1. Please note that the specific URLs for GSE255628 and GSE255643 datasets are not provided in the data availability statement.

o Figure legends:

1. Please note that the exact p values are not provided in the legends of figures 1e-i; 2a-d, h; 4d-e, g; 5h; 6c, g; EV 1c, g-j; EV 2a.
 2. Please indicate the statistical test used for data analysis in the legend of figure EV 5a.
 3. Please note that in figure EV 4a; there is a mismatch between the annotated p values in the figure legend and the annotated p values in the figure file that should be corrected.
 4. Please note that information related to n is missing in the legends of figures 2b-d; EV 1b.
 5. Please note that for heatmap present in figure 5e a numbered scale bar is not provided. This needs to be rectified.
- Correct order of manuscript sections to: Abstract, The Paper Explained, Introduction, Results, Discussion, Methods, Acknowledgements, Disclosure and competing interests statement, References, Figure legends, Tables and their legends, Expanded View Figure legends.
 - Add up to 5 keywords.

- Correct callout Supplementary reagents Table 1 to Reagents Table.

- Rename "Competing interests" to "Disclosure and competing interests statement". We updated our journal's competing interests policy in January 2022 and request authors to consider both actual and perceived competing interests. Please review the policy <https://www.embopress.org/competing-interests> and update your competing interests if necessary.

- Author contributions: Please remove it from the manuscript and specify author contributions in our submission system. CRediT has replaced the traditional author contributions section because it offers a systematic machine-readable author contributions format that allows for more effective research assessment. You are encouraged to use the free text boxes beneath each contributing author's name to add specific details on the author's contribution. More information is available in our guide to authors:

<https://www.embopress.org/page/journal/17574684/authorguide#authorshipguidelines>

- Remove "For more information".

- Correct the reference citation in the reference list. In the reference list, where there are more than 10 authors on a paper, 10 will be listed, followed by "et al.". Please check "Author Guidelines" for more information.

<https://www.embopress.org/page/journal/17574684/authorguide#referencesformat>

- Rename "Data and materials availability" to "Data availability". Please be aware that all deposited datasets should be made freely available upon acceptance, without restriction. Please check "Author Guidelines" for more information.

<https://www.embopress.org/page/journal/17574684/authorguide#availabilityofpublishedmaterial>

3) Reagent table: Please use the template provided in our author guidelines. More information on how to adhere to this format as well as downloadable templates (.docx) for the Reagents and Tools Table can be found in our author guidelines:

<https://www.embopress.org/page/journal/17574684/authorguide#structuredmethods>

4) Funding: Please merge it with "Acknowledgements".

5) Synopsis:

- Synopsis image: Please resize the image to 550 px-wide x (250-400)-px high and upload it as a high-resolution jpeg file.
- Please check your synopsis text and image before submission with your revised manuscript. Please be aware that in the proof stage minor corrections only are allowed (e.g., typos).

6) As part of the EMBO Publications transparent editorial process initiative (see our Editorial at

<http://embomolmed.embopress.org/content/2/9/329>), EMBO Molecular Medicine will publish online a Review Process File (RPF) to accompany accepted manuscripts. This file will be published in conjunction with your paper and will include the anonymous referee reports, your point-by-point response and all pertinent correspondence relating to the manuscript. Let us know whether you agree with the publication of the RPF and as here, if you want to remove or not any figures from it prior to publication. Please note that the Authors checklist will be published at the end of the RPF.

7) Please provide a point-by-point letter INCLUDING my comments as well as the reviewer's reports and your detailed responses (as Word file).

I look forward to reading a new revised version of your manuscript as soon as possible.

Yours sincerely,

Zeljko Durdevic

*** Instructions to submit your revised manuscript ***

To submit your manuscript, please follow this link:

<https://embomolmed.msubmit.net/cgi-bin/main.plex>

- 1) a .docx formatted version of the manuscript text (including Figure legends and tables)
- 2) Separate figure files*
- 3) supplemental information as Expanded View and/or Appendix. Please carefully check the authors guidelines for formatting Expanded view and Appendix figures and tables at <https://www.embopress.org/page/journal/17574684/authorguide#expandedview>
- 4) a letter INCLUDING the reviewer's reports and your detailed responses to their comments (as Word file).
- 5) The paper explained: EMBO Molecular Medicine articles are accompanied by a summary of the articles to emphasize the major findings in the paper and their medical implications for the non-specialist reader. Please provide a draft summary of your article highlighting
 - the medical issue you are addressing,
 - the results obtained and
 - their clinical impact.This may be edited to ensure that readers understand the significance and context of the research. Please refer to any of our published articles for an example.
- 6) For more information: There is space at the end of each article to list relevant web links for further consultation by our readers. Could you identify some relevant ones and provide such information as well? Some examples are patient associations, relevant databases, OMIM/proteins/genes links, author's websites, etc...
- 7) Author contributions: the contribution of every author must be detailed in a separate section.
- 8) EMBO Molecular Medicine now requires a complete author checklist (<https://www.embopress.org/page/journal/17574684/authorguide>) to be submitted with all revised manuscripts. Please use the checklist as guideline for the sort of information we need WITHIN the manuscript. The checklist should only be filled with page numbers where the information can be found. This is particularly important for animal reporting, antibody dilutions (missing) and exact values and n that should be indicated instead of a range.

9) Every published paper now includes a 'Synopsis' to further enhance discoverability. Synopses are displayed on the journal webpage and are freely accessible to all readers. They include a short stand first (maximum of 300 characters, including space) as well as 2-5 one sentence bullet points that summarise the paper. Please write the bullet points to summarise the key NEW findings. They should be designed to be complementary to the abstract - i.e. not repeat the same text. We encourage inclusion of key acronyms and quantitative information (maximum of 30 words / bullet point). Please use the passive voice. Please attach these in a separate file or send them by email, we will incorporate them accordingly.

You are also welcome to suggest a striking image or visual abstract to illustrate your article. If you do please provide a jpeg file 550 px-wide x 300-600px high.

10) A Conflict of Interest statement should be provided in the main text

11) Please note that we now mandate that all corresponding authors list an ORCID digital identifier. This takes <90 seconds to complete. We encourage all authors to supply an ORCID identifier, which will be linked to their name for unambiguous name identification.

Currently, our records indicate that the ORCID for your account is 0000-0001-9513-0286.

Link Not Available

12) Include a Reagents and Tools Table as part of the Methods section, which can be downloaded from our author guidelines (<https://www.embopress.org/page/journal/17574684/authorguide#structuredmethods>)

Photos 400-800 DPI

*Additional important information regarding figures and illustrations can be found at <https://bit.ly/EMBOPressFigurePreparationGuideline>. See also figure legend preparation guidelines: <https://www.embopress.org/page/journal/17574684/authorguide#figureformat>

***** Reviewer's comments *****

Referee #1 (Remarks for Author):

The authors have responded satisfactorily to my comments.

Referee #3 (Comments on Novelty/Model System for Author):

While more xenograft tests of patient derived models would have added to the value of the manuscript, the data and models as presented are adequate.

Referee #3 (Remarks for Author):

The authors have responded appropriately to all of the reviewers' comments including providing addition experimental data and extensive editing, including discussion of "limitations of the current study" in the Discussion section.

The authors addressed the minor editorial issues.

manuscript. Please be aware that in the proof stage minor corrections only are allowed (e.g., typos).

We have provided the image as 550 px wide x 400px high. The synopsis text has been checked.

6) As part of the EMBO Publications transparent editorial process initiative (see our Editorial at <http://embomolmed.embopress.org/content/2/9/329>), EMBO Molecular Medicine will publish online a Review Process File (RPF) to accompany accepted manuscripts. This file will be published in conjunction with your paper and will include the anonymous referee reports, your point-by-point response and all pertinent correspondence relating to the manuscript. Let us know whether you agree with the publication of the RPF and as here, if you want to remove or not any figures from it prior to publication. Please note that the Authors checklist will be published at the end of the RPF.

We agree with the publication of the RPF and will not remove any figures from it prior publication.

7) Please provide a point-by-point letter INCLUDING my comments as well as the reviewer's reports and your detailed responses (as Word file).

We provided a point-by-point letter to all the reviewers and editor comments.

29th Aug 2024

Dear Prof. Konstantinidou,

We are pleased to inform you that your manuscript is accepted for publication and is now being sent to our publisher to be included in the next available issue of EMBO Molecular Medicine.
